# Single-shot 20-fold expansion microscopy

**Shiwei Wang** [1,2,15], **Tay Won Shin**[1,3,15], **Harley B. Yoder II**[1,4,5], **Ryan B. McMillan** [6,7,8], **Hanquan Su**[6,7], **Yixi Liu**[1,5], **Chi Zhang**[1], **Kylie S. Leung** [1], **Peng Yin** [6,7], **Laura L. Kiessling** [2,9,10] ✉ & **Edward S. Boyden** [1,3,4,10,11,12,13,14] ✉

Expansion microscopy (ExM) is in increasingly widespread use throughout biology because its isotropic physical magnification enables nanoimaging on conventional microscopes. To date, ExM methods either expand specimens to a limited range (~4–10× linearly) or achieve larger expansion factors through iterating the expansion process a second time (~15–20× linearly). Here, we present an ExM protocol that achieves ~20× expansion (yielding <20-nm resolution on a conventional microscope) in a single expansion step, achieving the performance of iterative expansion with the simplicity of a single-shot protocol. This protocol, which we call 20ExM, supports postexpansion staining for brain tissue, which can facilitate biomolecular labeling. 20ExM may find utility in many areas of biological investigation requiring high-resolution imaging.

Identifying and locating biomolecules with nanoscale precision in intact cells and tissues is key to understanding their roles in such biological systems. Expansion microscopy (ExM) provides a robust, simple and affordable solution because its isotropic physical magnification enables nanoscale-resolution imaging of preserved cells and tissues on conventional microscopes[1,2]. In ExM, a dense mesh of swellable hydrogel is formed throughout preserved biological specimens, with biomolecules and/or fluorescent tags covalently anchored to the polymer network. After the embedded specimens are chemically softened and the hydrogel is immersed in water, the polymer network expands isotropically while preserving the relative spatial organization of the anchored molecules. Previous ExM methods either expanded specimens to a limited range in one shot (~4–10× linearly)[3–7] or achieved higher expansion factors through re-embedding the first gel in a second hydrogel and then iterating the expansion process again (~15–20× linear expansion total)[8–10]. Many nanoscale biological features, such as the hollow structure of microtubules and the nanocolumnar alignment of synaptic proteins, have been visualized via such iterative expansion protocols, which involve multiple processing steps[8–10]. In one of these protocols, expansion revealing (ExR)[10], fluorescent antibodies are delivered to brain tissue after iterative expansion; by separating densely packed proteins from one another before antibody staining, antibodies attain better access to epitopes, in some cases converting virtually invisible molecular targets into visible ones.

Here, we report an ExM protocol that achieves the resolution of iterative expansion protocols (<20-nm resolution) with the simplicity of one-shot protocols, achieving ~20× expansion of cell cultures and tissues in a single expansion step and supporting postexpansion staining of biomolecules for brain tissue. In one round of expansion, this protocol, which we call 20ExM, reveals hollow microtubule structures in cultured cells and synaptic nanocolumns in mouse somatosensory cortex on a conventional confocal microscope. We anticipate 20ExM to find broad utility in biology due to its high performance and simplicity.

## Results

### Development of a state-of-the-art superabsorbent hydrogel
We sought to develop a superabsorbent hydrogel that could achieve expansion factors comparable with iterative expansion protocols

[1]McGovern Institute for Brain Research, Massachusetts Institute of Technology, Cambridge, MA, USA. [2]Department of Chemistry, Massachusetts Institute of Technology, Cambridge, MA, USA. [3]Media Arts and Sciences, Massachusetts Institute of Technology, Cambridge, MA, USA. [4]Department of Brain and Cognitive Sciences, Massachusetts Institute of Technology, Cambridge, MA, USA. [5]Department of Electrical Engineering and Computer Science, Massachusetts Institute of Technology, Cambridge, MA, USA. [6]Wyss Institute for Biologically Inspired Engineering, Harvard University, Boston, MA, USA. [7]Department of Systems Biology, Harvard Medical School, Boston, MA, USA. [8]Biophysics PhD Program, Harvard University, Cambridge, MA, USA. [9]Broad Institute of MIT and Harvard, Cambridge, MA, USA. [10]Koch Institute, Massachusetts Institute of Technology, Cambridge, MA, USA. [11]Department of Biological Engineering, Massachusetts Institute of Technology, Cambridge, MA, USA. [12]Center for Neurobiological Engineering, Massachusetts Institute of Technology, Cambridge, MA, USA. [13]K. Lisa Yang Center for Bionics, Massachusetts Institute of Technology, Cambridge, MA, USA. [14]Howard Hughes Medical Institute, Cambridge, MA, USA. [15]These authors contributed equally: Shiwei Wang, Tay Won Shin. ✉e-mail: kiesslin@mit.edu; edboyden@mit.edu

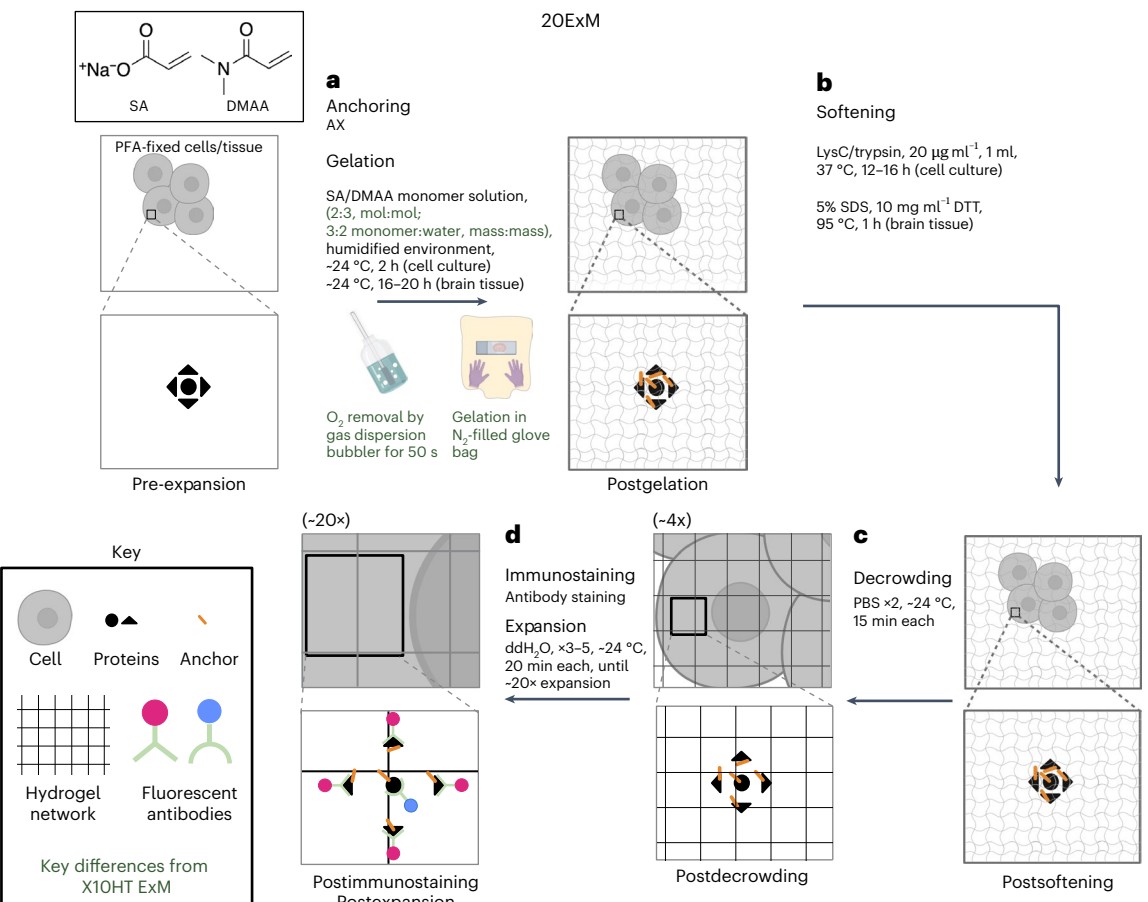

**Fig. 1 | Single-shot 20ExM. a–d**, Workflow for expanding cell culture and tissue samples ~20-fold with only one gelation step. Key differences from the published X10HT protocol (ref. 4) are shown in green text; PFA, paraformaldehyde; AX, *N*-acryloxysuccinimide; DTT, dithiothreitol; PBS, phosphate-buffered saline; ddH₂O, double-distilled water. For steps after decrowding (**c**), the linear expansion factor of the hydrogel–specimen composite is shown in parentheses above the schematic of the step. **a**, Cell culture or tissue samples are treated to attach gel-anchorable groups to proteins. The sample is then permeated with monomer solution and incubated to form a superabsorbent polyacrylate hydrogel. **b**, Samples are incubated in a softening buffer to denature proteins. **c**, Softened samples are washed in a buffer to partially expand them. **d**, Samples are stained with antibodies and fully expanded by immersion in water.

in a single expansion step. Some previous studies reached higher expansion factors by reducing cross-linker concentration in bis-acrylamide-cross-linked hydrogels; achieving higher expansion factors with this strategy can lead to lower gel structural integrity, and thus expansion factors beyond ~10× have not been achieved[1,6,7]. To address this limitation, a polymer with exceptional mechanical properties is needed. We chose to optimize a hydrogel composed of *N*,*N*-dimethylacrylamide (DMAA) and sodium acrylate (SA), reagents that are known to form mechanically robust and elastic hydrogels due to the unique self-cross-linking chemistry of DMAA (Extended Data Fig. 1)[11]. Versions of this hydrogel have been used to create a one-shot 10× ExM protocol, called the X10 protocol[3,4].

We tested whether these reagents could afford gels with higher expansion factors. Starting from the X10 protocol, we increased the SA:DMAA molar ratio from 1:4 to 2:3 and the monomer (SA and DMAA):water mass ratio from 1:2 to 3:2 (Fig. 1a) to reach expansion factors beyond tenfold. However, we observed large batch-to-batch variation in expansion factor and gel mechanical properties. We hypothesized that this variation was due to varying oxygen concentration in the monomer solution. Compared to bis-acrylamide-cross-linked hydrogels, DMAA–SA hydrogels are especially sensitive to oxygen because their polymerization relies on a radical-dependent mechanism to self-cross-link (Extended Data Fig. 1)[11]. As oxygen can react rapidly with the intermediate radicals, its concentration during gelation can substantially affect the cross-linking density and, subsequently, gel

properties. Previous DMAA–SA-based 10× ExM methods required bubbling nitrogen gas through a needle for 40 min before gelation to displace dissolved oxygen[3,5]. However, even with prolonged nitrogen bubbling (up to 3 h), our optimized gel formula still suffered from batch-to-batch variation.

We reasoned that performing gelation in an oxygen-controlled environment could reduce this batch-to-batch variation. To this end, we deoxygenated the gelation solution by flowing nitrogen through a gas dispersion tube immersed in the solution for 50 s. This procedure breaks the nitrogen flow into tiny bubbles, which streamlines oxygen displacement. We then moved the gelation solution into a countertop glove bag connected to a compressed nitrogen gas cylinder (a simple and inexpensive means to manipulate objects in an oxygen-depleted environment, thus enabling processes such as ExM to be performed without requiring specialized equipment not found in a typical biology laboratory). The resulting setup (depicted in Extended Data Figs. 2 and 3, with a step-by-step protocol in Supplementary Note 1) substantially improved reproducibility of gels exhibiting high expansion factors (Extended Data Fig. 4a). The improved removal of oxygen supported by the glove bag was essential for our optimized reaction conditions to consistently afford materials that gelated. When oxygen was present, the reagents would sometimes simply fail to yield gels. With this new protocol, we found that the expansion factor, now reliable across batches made under the same conditions, would vary systematically with gelation time. We stopped gelation (by placing the gel

in double-distilled water at room temperature) after different time periods. For the same gel formula, the gel would expand 16× when gelation was stopped after 1 h; 2 h afforded 13× expansion, and 16 h afforded 8× expansion (Extended Data Fig. 4b). Transitioning from pure gels to specimen–gel composites, we found that the presence of a biological specimen altered the polymerization kinetics: the same gel formula, with a cell or tissue embedded, required a longer time to gelate for a given targeted expansion factor than a pure gel. We also tuned initiator concentration and gelation time for tissues to allow monomers to fully permeate specimens. Through optimizing initiator concentration (7.7 μM for cell culture and 1.6 μM for brain tissue) and gelation time (2 h for cell culture and 16–20 h for brain tissue), we reliably achieved expansion factors of 21.50 ± 1.70 (mean ± s.d. used throughout unless otherwise indicated; $n = 8$ cells from four culture batches) for cell culture and 18.44 ± 0.33 for brain tissue ($n = 2$ brain slices from one mouse; Fig. 1 and Supplementary Table 2). These expansion factors were consistent whether measured via physical gel size or utilization of biological landmarks, and were identical in all directions ($x$, $y$ and $z$; Supplementary Note 2 and Supplementary Data). Gelation time is a critical parameter in ensuring reproducible expansion (Supplementary Note 3). Expanded gels were stable for periods of many hours after expansion (as long as we examined), as long as humidity was maintained (Supplementary Note 4).

### Validation of 20ExM resolution

To validate the resolution of 20ExM, we stained microtubules in cell culture using pre-expansion primary and secondary antibody staining, performed softening via proteolysis with LysC/trypsin digestion and performed postexpansion further staining (for example, with a tertiary antibody), expansion and imaging, similar to the iterative ExM (iExM) protocol[8]. We visualized the hollow structure of microtubules, which has been used as a benchmark for resolution in various studies (Supplementary Table 1)[3,4,6–9,12,13]. iExM affords ~×20 physical magnification with three rounds of gelation to yield an average distance between microtubule sidewall peaks of 58.7 ± 10.3 nm on a conventional confocal microscope, as expected for this antibody staining and signal amplification scheme. With 20ExM, we saw hollow microtubule structures with an appearance consistent with that observed in previous studies such as the iExM study, with an average sidewall peak distance of 62.1 ± 8.8 nm, indistinguishable from that yielded by iExM (Fig. 2a–c). The standard deviation of 8.8 nm, in particular, could be regarded as an upper bound (because it includes any real biological variability in microtubule thickness) on the nanoscale error introduced by 20ExM and is similar to that observed for iExM above (see Supplementary Note 5 for further discussion of microtubule diameter and how our measurements compare to those observed with other technologies). As reported in iExM, we also observed circular cross-sections of microtubules when they happened to be perfectly orthogonal to the imaging plane (Fig. 2d–f). Thus, the hollow structure of microtubules was easily resolved and characterized (with quality on par with state-of-the-art iterative protocols) via the single-step 20ExM protocol.

To more quantitatively evaluate the resolution that 20ExM provides, we used block-wise Fourier ring correlation (FRC) resolution analysis[14], which measures the resolution of an image by evaluating a normalized cross-correlation histogram measure in the frequency domain between two images that captured the same region under the same imaging conditions (Fig. 2i). We performed FRC analysis on 34 image pairs of microtubules from two biological replicates. We observed an effective resolution of 17.9 ± 1.3 nm (median = 18.7; Fig. 2j), comparable to the highest reported resolution of iterative expansion protocols (Supplementary Table 1). This analysis was robust to the levels of noise we estimated to occur in our images (Supplementary Note 6). To evaluate the distortion of 20ExM over nanoscale distances, we analyzed, as in the iExM paper[8], the variation of microtubule diameter along 185-nm distances randomly selected along the long axis of

imaged microtubules. The estimated distortion was found to be 8.8 nm (Fig. 2c), indistinguishable from the published distortion measure of iExM of 10.3 nm (ref. 8).

### 20ExM reveals *trans*-synaptic nanoarchitecture in the mouse brain

To demonstrate 20ExM's utility in brain tissue, we imaged synaptic nanocolumns, which were visualized previously with STORM and ExR[10,15]. We first evaluated the macroscopic distortion of ExM in expanded brain slices using standard ExM distortion analysis[7,8,16,17] methods, which calculate a root mean square (r.m.s.) alignment error from the deformation vector field obtained by comparing pre- and postexpansion images of the same field of view. We obtained low distortion comparable to previous ExM protocols in both $x$ and $y$ (Fig. 2g,h) and in $z$ (see Supplementary Note 8). We applied 20ExM to paraformaldehyde-fixed adult mouse brain slices, followed by postexpansion staining against RIM1/2 and PSD95, presynaptic and postsynaptic scaffolding proteins among those examined in the ExR study. We used postexpansion staining because it has been shown to be capable of revealing otherwise unseen proteins through decrowding densely packed regions for better antibody access. To compare our results with ExR, we imaged in the same region investigated in the earlier study, specifically layers 2 and 3 of the somatosensory cortex (Fig. 3a). We observed a juxtaposition of RIM1/2 and PSD95 scaffolds with 20ExM, similar to what was observed with ExR (Fig. 3b). We then performed the three-dimensional autocorrelation $g_a(r)$ analysis used in previous studies[10,15,18] to look for inhomogeneous distributions of proteins. A more heterogeneous distribution within a synapse will result in higher $g_a(r)$, and the distance at which $g_a(r)$ flattens can be used to estimate the size of each internal cluster, sometimes termed a nanodomain. For both RIM1/2 and PSD95, we observed nanodomains with sizes of ~50–70 nm (Fig. 3c,d), consistent with previous reports. To analyze the spatial alignment of the two distributions, we performed protein enrichment analysis, which measures volume-averaged intensity of one channel as a function of distance from the peak intensity of another channel (see Methods for details). We evaluated protein enrichment for RIM1/2 relative to the PSD95 peak (Fig. 3e) and PSD95 relative to the RIM1/2 peak (Fig. 3f). Both intensities flattened around ~20–25 nm away from peak, indicating precise alignment between presynaptic RIM1/2 and postsynaptic PSD95, consistent with previous reports. These results demonstrate that 20ExM can visualize synaptic nanoarchitecture that had been previously documented with confocal imaging of iteratively expanded samples (for example, via ExR) or with single-molecule localization microscopy (for example, STORM) imaging. We analyzed the signal-to-noise ratio (SNR) of our signals and found them comparable to those obtained with ExR, perhaps because in both cases, postexpansion antibody staining permits much higher levels of staining than pre-expansion antibody staining (Supplementary Note 6).

### Applications of 20ExM

To explore the utility of 20ExM, we performed 20ExM to image more organelles and tissue types. We explored visualizing nuclear pore complexes (NPCs), which have been imaged with various ExM methods[6,19,20]. These papers used a variety of fixation and extraction methods, ranging from permeabilization with detergent before fixation to using paraformaldehyde of varied concentrations or methanol cryofixation, with some methods extracting nuclei from intact cells before staining. Because our goal was to validate 20ExM as would be experienced in everyday biology rather than to study NPCs per se, we simply used standard 4% paraformaldehyde to fix intact cells in which the nuclear pore protein NUP96 was fused to the fluorescent protein mNeonGreen[21,22] and stained with anti-mNeonGreen. We performed 20ExM and imaged NPCs on the top and bottom of the nuclei, which are tangential to the imaging plane, to facilitate observation of the shape of the nuclear pore in the imaging plane. We observed the ring

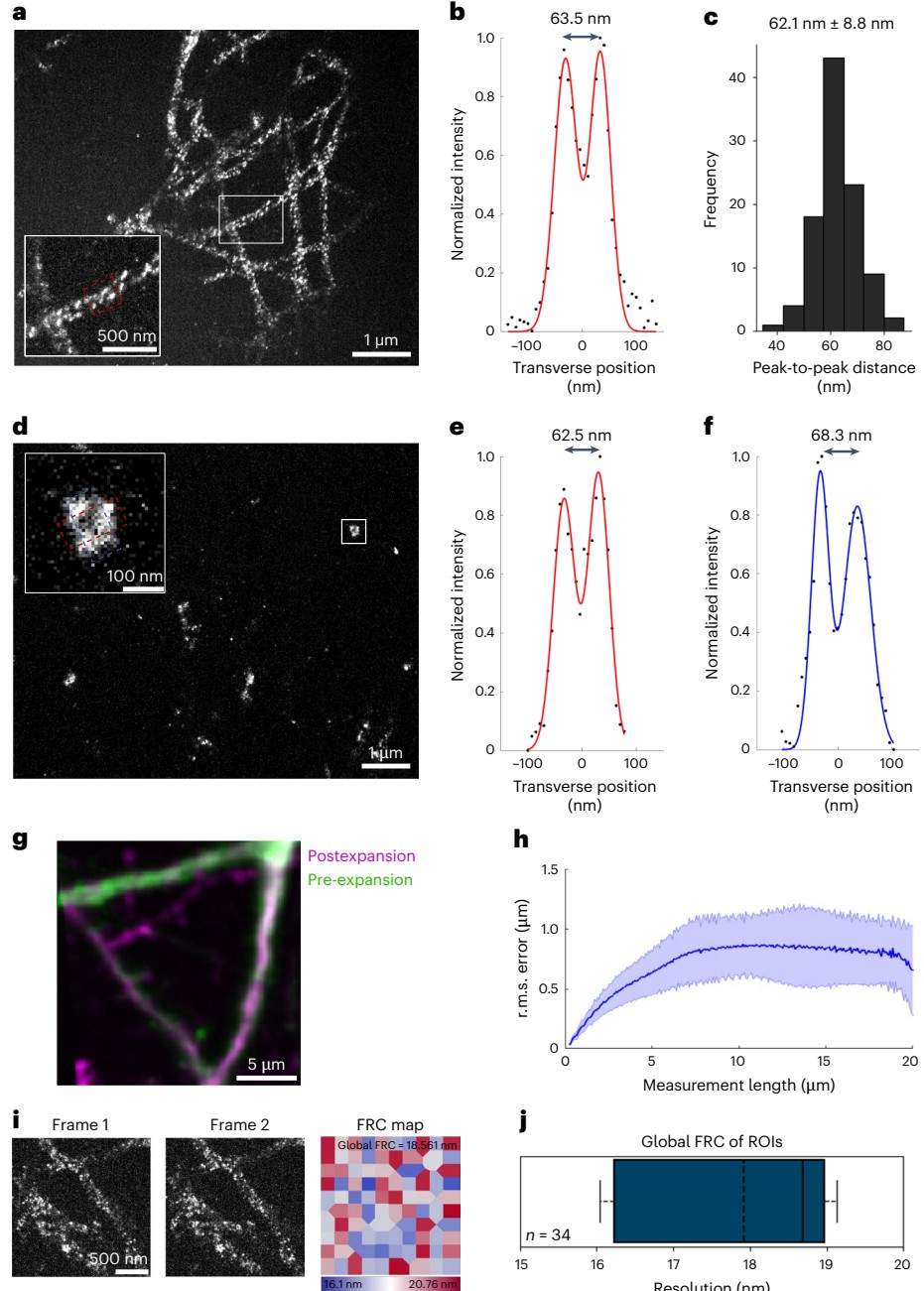

**Fig. 2 | Validation of the nanoscale precision of 20ExM. a**, Confocal image (maximum intensity projection from one representative experiment of three culture batches) of expanded HEK293 cells with pre-expansion microtubule staining. The inset shows a magnified view of the white boxed region. Brightness and contrast settings were set using Fiji's autoscaling function. Quantitative analysis in **b** and **c** was conducted on raw image data. **b**, Transverse profile of microtubules in the red dotted boxed region of the inset in **a** after averaging down the long axis of the box and then normalizing to peak value (black dots), with superimposed fit with a sum of two Gaussians (red lines). **c**, Population data for peak-to-peak distances of 100 microtubule segments (mean ± s.d. from 21 cells from three culture batches). **d**, Confocal image (single *xy* plane from one representative experiment of three culture batches) of expanded HEK293 cells with pre-expansion microtubule staining. The inset shows a magnified view of the white boxed region, highlighting the microtubule circular cross-section. Brightness and contrast settings were set using Fiji's autoscaling function. Quantitative analysis in **e** and **f** was conducted on raw image data. **e**, As in **b** but for the red dotted box in the inset of **d**. **f**, As in **b** but for the blue dotted box in the inset of **d**. **g**, Nonrigidly registered pre-expansion ×40 magnification confocal image (green) and postexpansion ×4 magnification confocal image (magenta) of

the same region in the same Thy1–yellow fluorescent protein mouse brain slice (from one representative experiment of two brain slices from one mouse). **h**, r.m.s. measurement error as a function of measurement length of data acquired as in **g** (blue line, mean; shaded area, ±1 s.d.; *n* = 6 areas from two brain slices from one mouse). **i**, To measure resolution, we used block-wise FRC resolution analysis[14]. The method requires more than one independent image of the same region for noise realization. Left and middle, two independent confocal images (single *xy* plane) of expanded HEK293 cells with pre-expansion microtubule staining, showing the same region of interest under the same imaging conditions. Right, local mapping of FRC resolution values. A global FRC resolution is calculated by averaging FRC resolution values across all blocks. **j**, Box plot of global FRC resolution calculated for *n* = 34 regions of interest from two culture batches (black vertical line, median; dotted vertical line, mean; leftmost edge of the box, first quartile; rightmost edge of the box, third quartile; left dotted line extended from the box, first quartile minus 1.5× the interquartile range; right dotted line extended from the box, third quartile plus 1.5× the interquartile range). Scale bars are provided in biological units (that is, physical size divided by expansion factor) for all images; ROIs, regions of interest.

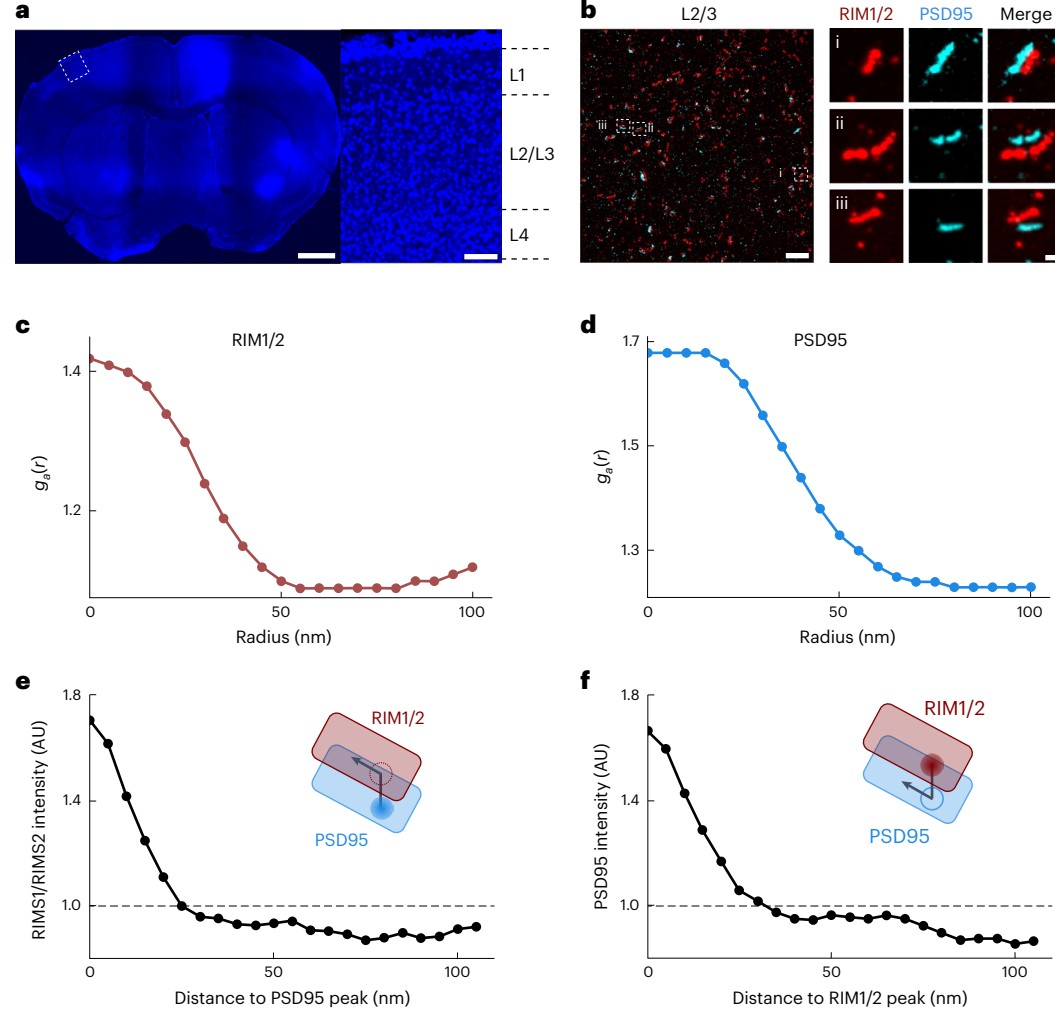

**Fig. 3 | 20ExM reveals synaptic nanoarchitecture in mouse brain tissue.**
**a**, Confocal image of a DAPI-stained mouse brain slice (left) and zoomed-in view (right) of the white dotted boxed region showing layers 1–4 of the somatosensory cortex (from one representative experiment of two brain slices from one mouse). **b**, Maximum z intensity-projected confocal image of layers 2 and 3 of the mouse somatosensory cortex after performing 20ExM and postexpansion immunostaining with antibodies to RIM1/2 (red) and PSD95 (cyan). Left, low-magnification image. Right, zoomed-in images of the three white dotted boxes (**i**–**iii**) with separate channels for each antibody along with the merged image. The image shown is from a representative experiment using four brain slices from two mice. Brightness and contrast settings were first set by Fiji's autoscaling function and then manually adjusted to improve contrast and highlight the boundary of the synapses; quantitative analysis in **c**–**f** was conducted on raw image data. **c**,**d**, Autocorrelation analysis, as described in refs. 10,15, for RIM1/2 (**c**) and PSD95 (**d**; n = 90 synapses from four brain slices from two

mice). Autocorrelation analysis examines the protein distribution. A uniform distribution would be predicted if baseline $g_a(r)$ values are observed at all radii, whereas a nonuniform distribution with regions of high local intensity would be predicted if high $g_a(r)$ values are observed at short radii and decay as the radius is increased. **e**,**f**, Enrichment analysis that calculates the average molecular density for RIM1/2 to PSD95 peak (**e**) and PDS95 to RIM1/2 peak (**f**; n = 90 synapses from four brain slices from two mice). Enrichment values above 1 represent regions of high local intensity in the measured channel, so the enrichment profiles in **e** and **f** suggest that the peak of the reference channel closely aligns with the regions of high intensity in the measured channel for both comparisons. Therefore, this suggests that enriched regions of RIM1/2 and PSD95 are aligned in nanoscale precision with each other, consistent with previous studies[10,15]. Scale bars are provided in biological units: 1,000 μm (left) and 100 μm (right; **a**), 1 μm (left) and 100 nm (right, **i**–**iii**; **b**); AU, arbitrary units.

structure of individual NPCs (Extended Data Fig. 5a). We then manually picked NPCs with at least four visible corners in top view and measured the radius to be 55.4 ± 8.9 nm (median = 58.7; Extended Data Fig. 5b), consistent with the expected radius of 53.5 nm based on the previously reported cryoelectron microscopy structure[19] and previous ExM reports[6,20]. Furthermore, the standard deviation of 8.9 nm serves as an expansion error upper bound and was comparable to that measured using microtubule diameter, above, for both 20ExM and iExM (see the discussion above). 20ExM clearly resolved individual corners within NPC rings, which are around 42 nm apart from each other based on previous cryo-EM data and have been visualized by dSTORM[19] and the iterative expansion method iU-ExM[20]. We counted the number of

corners per NPC using a previously reported algorithm[20] and observed a similar distribution of numbers of corners per NPC as previous studies (Extended Data Fig. 5c). We then measured the distance between adjacent corners to be 48.6 ± 12.8 nm (median = 48.9; Extended Data Fig. 5d), consistent with the expected distance of 42 nm and with an expansion error upper bound (the aforementioned standard deviation of 12.8 nm) comparable to that measured using microtubule diameter or NPC radius (see Supplementary Note 7 for further discussion).

We visualized the outer mitochondrial membrane by immunostaining for the outer membrane protein TOM20 and observed the hollow structure of mitochondria (Extended Data Fig. 5e), consistent with previous STORM images[23].

We tested kidney and spleen sections with the standard 20ExM tissue protocol. We found that with standard sodium dodecyl sulfate (SDS) softening, gels containing kidney and spleen tissue became distorted and folded. This is consistent with our previous observations, where tissues that are more fibrous than brain may require stronger softening than achieved with heat and detergent alone[24]. With a stronger digestion protocol, LysC/trypsin proteinase digestion[25], appropriate for pre-expansion staining, both kidney and spleen reached 16.5-fold (±0.4) expansion (Extended Data Fig. 5f–h), slightly less than brain tissue expanded under its corresponding 20ExM protocol but still higher than achieved with previous single-shot protocols. Thus, we recommend the standard gelation condition for all tissues, at least as a starting point (very complex tissues like bone and cartilage or very large samples like entire mammalian brains may of course require further tuning), but tissues with challenging mechanical properties may require harsher softening methods than heat/detergent treatment, such as enzymatic methods, many of which have already been validated and published by us and others.

## Discussion

20ExM achieves a resolution comparable to iterative expansion methods (<20 nm) with a single expansion step. As demonstrated in both cell culture and tissue specimens, 20ExM can be immediately deployed in a wide variety of experimental contexts where high resolution and single-step simplicity are desired. 20ExM could, in principle, be used to simplify and/or enhance the resolution of other expansion-based technologies, such as in situ RNA detection and sequencing[26–29], genome imaging[30–32], multiplexed proteomics[33–36] and lipid and glycan staining[7,25,37–42].

Due to 20ExM's high expansion factor, which dilutes signal density, signal amplification is very useful. For samples with postexpansion antibody staining, primary and fluorescent secondary antibodies can afford sufficient signal intensity. For cell cultures and tissues stained with primary and secondary antibodies before expansion, we used fluorescent tertiary antibody staining (targeting the secondary antibody) to achieve enhanced signal intensity (see Supplementary Note 1 for more information). Alternatively, previously published signal amplification methods could, in principle, be used, including hybridization chain reaction and rolling circle amplification, which, as modular DNA-based methods, have easily been incorporated into ExM protocols by multiple groups.

20ExM, in the form presented here, does not universally support postexpansion antibody staining of cell culture or mechanically challenging tissues such as kidney and spleen due to the limitations of SDS softening. For example, for cell cultures fixed with 3% paraformaldehyde and 0.1% glutaraldehyde, which is required to preserve the ultrastructure of microtubules[12] and mitochondria[23], we found that SDS softening at 95 °C did not enable full isotropic expansion of these nanostructures. For tissues with challenging mechanical properties, such as kidney and spleen, SDS softening led to gel distortion. Novel softening methods that are harsher than standard SDS softening but that preserve protein epitopes, such as SDS softening over prolonged timescales (for example, 80 h) or at higher temperatures (for example, 121 °C), as described in the Magnify[7] and dExPath[24] papers, may be useful in the future for creating forms of 20ExM that enable isotropic expansion for postexpansion staining of cell cultures, nonbrain mouse tissues and potentially human clinical tissues.

## Online content

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

## Methods

### Cell culture preparation

HEK293 cells (Thermo Fisher, R70007) were cultured in 16-well chambered coverglasses (Grace Bio-Labs, 112359) with DMEM supplemented with 1% penicillin–streptomycin, and the cells were incubated at 37 °C in 5% $CO_2$ to reach ~20% confluency. NUP96::Neon-AID DLD-1 cells (gifted by T. Schwartz, Massachusetts Institute of Technology (MIT)) were cultured in 24-well glass-bottom plates (Cellvis, P24-1.5H-N), with a 12-mm number 2 round glass coverslip at the bottom of each well, in DMEM supplemented with 1% penicillin–streptomycin, and the cells were incubated at 37 °C in 5% $CO_2$ to reach ~20–40% confluency.

Microtubule staining was performed following previously reported protocols[8,43]. All of the following steps were conducted at room temperature (~24 °C), unless otherwise noted. Cells were incubated in extraction buffer (0.5% (wt/vol) Triton X-100, 0.1 M 1,4-piperazinediethanesulfonic acid, 1 mM ethylene glycol-bis(2-aminoethylether)-$N,N,N',N'$-tetraacetic acid and 1 mM magnesium chloride (pH 7); 100 µl per well) for 1 min and fixed in tubulin fixation solution (3% formaldehyde, 0.1% glutaraldehyde and 1× PBS; 200 µl per well) for 10 min, followed by incubation in reduction solution (0.1% (wt/vol) sodium borohydride in 1× PBS; 200 µl per well) for 7 min and washing with quenching solution (100 mM glycine in 1× PBS; 200 µl per well) for 10 min. Cells were incubated in blocking buffer (Active Motif, 15252; 60 µl per well) for 2 h and then with rabbit anti-β-tubulin in staining buffer (Active Motif, 15253; 1:100 dilution, 60 µl per well) for 2 h. Samples were then washed in washing buffer (Active Motif, 15254; 100 µl per well) three times for 5 min each. Primary antibody staining and washes were then repeated under the same conditions. Cells were incubated with anti-rabbit secondary antibody diluted in staining buffer (1:100 dilution, 60 µl per well) for 2 h and washed in washing buffer (100 µl per well) three times for 10 min each. Secondary antibody staining and washes were then repeated under the same conditions.

TOM20 mitochondria staining was performed using HEK293 cells and previously reported protocols[23]. Cells were fixed in fixation solution (3% formaldehyde, 0.1% glutaraldehyde and 1× PBS; 200 µl per well) for 10 min, incubated in reduction solution (0.1% (wt/vol) sodium borohydride in 1× PBS; 200 µl per well) for 7 min and washed with quenching solution (100 mM glycine in 1× PBS; 200 µl per well) for 10 min. Cells were incubated in blocking buffer (Active Motif, 15252; 60 µl per well) for 2 h and then with rabbit anti-TOM20 diluted in staining buffer (Active Motif, 15253; 1:100 dilution, 60 µl per well) for 2 h. Samples were washed in washing buffer (Active Motif, 15254; 100 µl per well) three times for 5 min each, incubated with anti-rabbit secondary antibody in staining buffer (1:100 dilution, 60 µl per well) for 2 h and washed three times for 10 min each in washing buffer (100 µl per well).

NPC staining was performed on NUP96::Neon-AID DLD-1 cells. Cells were fixed in fixation solution (4% formaldehyde and 1× PBS; 1 ml per well) for 10 min and incubated with quenching solution (100 mM glycine in 1× PBS; 1 ml per well) for 10 min. Cells were incubated in blocking buffer (Active Motif, 15252; 300 µl per well) for 2 h and then with rabbit anti-mNeonGreen diluted in staining buffer (Active Motif, 15253; 1:100 dilution, 300 µl per well) for 2 h. Samples were washed in washing buffer (Active Motif, 15254; 500 µl per well) three times for 5 min each, incubated with anti-mouse secondary antibody diluted in staining buffer (1:100 dilution, 300 µl per well) for 2 h and washed in washing buffer (500 µl per well) three times for 10 min each.

All cells were incubated in AX solution (N-acryloxysuccinimide; Thermo Scientific, 400300010; dilution of 10 mg ml$^{-1}$ DMSO stock in 1× PBS, 1:2,000; 60 µl per well for 16-well chambered coverglass or 300 µl per well for 24-well glass-bottom plates) at room temperature (~24 °C) overnight (12–20 h). The cells were then washed in 1× PBS for 10 min.

### Tissue preparation

All procedures involving mice (Thy1-YFP-H, 6–8 weeks of age from The Jackson Laboratory, used without regard to sex and maintained under standard housing conditions on a 12-h light/12-h dark cycle at an ambient temperature and humidity) were performed in accordance with the US National Institutes of Health Guide for the Care and Use of Laboratory Animals and were approved by the MIT Committee on Animal Care. Mice were deeply anesthetized with isoflurane and perfused with 30 ml of 1× PBS, followed by 30 ml of 4 °C fixative solution (4% paraformaldehyde in 1× PBS). Brains, kidneys and spleens were then removed and stored in the same fixative at 4 °C overnight (12–18 h). Fixed brains, kidneys and spleens were transferred to 100 mM glycine at 4 °C for 6 h and sectioned to 50-µm-thick coronal slices with a vibrating microtome (Leica, VT1000S). The slices were stored in 1× PBS at 4 °C.

Before expansion, each brain slice was incubated in AX solution (Thermo Scientific, 400300010; dilution of 10 mg ml$^{-1}$ DMSO stock in 100 mM MES and 150 mM NaCl (pH 6) buffer, 1:200, 1 ml) at 4 °C overnight (12–20 h). The slices were then washed with 1 ml of 1× PBS for 10 min at room temperature (~24 °C).

Before expansion, kidney and spleen slices were microdissected into ~1 mm × 1 mm sections. Each section was incubated in AX solution (Thermo Scientific, 400300010; dilution of 10 mg ml$^{-1}$ DMSO stock in 100 mM MES and 150 mM NaCl (pH 6) buffer, 1:200, 50 µl) at 4 °C overnight (12–20 h). The sections were then washed with 50 µl of 1× PBS for 10 min at room temperature (~24 °C).

### Expansion of cell culture and tissue slices

See Supplementary Note 1 for a step-by-step protocol.

To generate hydrophobic glass for the gelation chamber, glass slides and coverslips were immersed in 0.2% (vol/vol) trichloro-(octadecyl)silane (Fisher Scientific, AC147400250) in hexane for 90 s. The coverslips were rinsed with 70% isopropanol and double-distilled water sequentially. The glass was dried at 37 °C and wiped with a dry Kimwipe to clear residual white solid. Parafilm strips were cut to ~4.5 cm × 0.2 cm and were wrapped around the glass slide to construct a gelation chamber with a 0.4-cm gap for cell culture or 0.1-cm gap for brain tissue (Extended Data Fig. 3a,e).

The gelation solution was prepared by dissolving 0.522 g of SA (AK Scientific, R624) in 1 ml of acidified Tris buffer (10% (vol/vol) 1 M Tris-HCl (pH 8) buffer and 20% (vol/vol) 1.2 M HCl in double-distilled water), followed by the addition of 7.5 µl of 10% (vol/vol) tetramethylethylenediamine (Sigma, T7024) in double-distilled water and 900 µl of DMAA (Sigma, 274135). The mixture was vortexed, yielding a colorless and noncloudy solution. The gelation solution was then placed on ice and bubbled with a gas dispersion tube (ChemGlass, CG-203-04) connected to a compressed nitrogen cylinder tank at a minimal flow rate for 50 s (Extended Data Fig. 2a). The gelation solution was removed from ice and allowed to return to room temperature (~24 °C). All of the following gelation steps were conducted at room temperature. Gelation solution, initiator solution (potassium persulfate, 45 mg ml$^{-1}$ in double-distilled water), cell culture or brain tissue, pipettes (P1000, P200 and P20), pipette tips, humidified chamber, hydrophobic glass slides and coverslips, tweezers, a transfer pipette and empty 1.5-ml centrifuge tubes were moved into a glove bag (GlasCol, 108D X-17-17HG) connected to a compressed nitrogen cylinder tank (Extended Data Fig. 2b).

For AX-treated cell cultures, the coverglass from the cell culture well was separated using a coverglass removal tool (Grace Bio-Lab, 103259; Extended Data Fig. 3d). Parafilm strips on the glass slide were adjusted to match with the positions of the wells to be expanded. The remaining rubber was carefully removed from the coverglass with tweezers. The coverglass was placed on top of the parafilm strips with the cells facing up, and 1× PBS was added to keep the cell culture hydrated (Extended Data Fig. 3e). All samples, solutions and tools were moved into the glove bag. The glove bag was purged three times by repeatedly

filling the bag with nitrogen and pushing down on the bag to expel most of the accumulated gas. The bag was then sealed and filled with nitrogen. If, in rare occurrences, the bag was leaky and slowly deflated when sealed without nitrogen flow added, a small flow of nitrogen was provided to keep the bag inflated. Inside the glove bag, 20 µl of initiator solution was added to 411 µl of gelation solution in a 1.5-ml centrifuge tube. The tube was flipped upside down five times to mix. The 1× PBS was removed from the cell culture coverglass with a transfer pipette, and 50 µl of the activated gelation solution was added to each well of cell culture. The coverglass was then flipped upside down with tweezers and placed on the parafilm strips to form the gelation chamber (Extended Data Fig. 3f). The gelation chamber was placed in an airtight humidified chamber, taken out of the glove bag and incubated at room temperature (~24 °C) in the dark for 2 h. After incubation, the portion of gel containing cell culture was cut out from the chamber and incubated in digestion buffer (20 µg of LysC/trypsin proteinase in 1 ml of 100 µM Tris-HCl (pH 8) buffer per gel) at 37 °C overnight (12–16 h). Digested gels were washed in PBS two times for 15 min each before proceeding to immunostaining.

For AX-treated mouse brain slices, the brain slices were microdissected to acquire somatosensory cortex as previously reported[10]. All microdissected brain, kidney or spleen slices were placed on a glass slide immersed in 1× PBS (Extended Data Fig. 3a,b). All samples, solutions and tools were moved into the nitrogen gas-filled glove bag, followed by three purges as described above. Inside the glove bag, 4 µl of initiator solution was added to 411 µl of gelation solution in a 1.5-ml centrifuge tube. Please note that we added 4 µl of initiator solution for tissue but 20 µl for cell culture. We optimized initiator concentration and gelation time for the tissue protocol to ensure ample time for monomer solution to diffuse into the brain slice. The tube was flipped upside down five times for mixing. The 1× PBS immersing the tissue was removed with a transfer pipette, and 50 µl of the solution was added to incubate the tissue for 15 min in a humidified chamber; the gelation chamber was then constructed by placing a coverslip on top (Extended Data Fig. 3c). The gelation chamber was placed in an airtight humidified chamber, taken out of the glove bag and incubated at room temperature (~24 °C) in the dark overnight (16–20 h).

After incubation, a portion of the gel containing the brain tissue was cut out from the chamber and incubated in denaturation buffer (1 ml; 5% (vol/vol) SDS, 200 mM NaCl, 50 mM Tris (pH 8) and 10 mg ml$^{-1}$ DTT) for 1 h at 95 °C. Denatured gels were washed in 1× PBS two times for 15 min each before proceeding to immunostaining.

The gel containing kidney or spleen tissue was cut out from the chamber and incubated in digestion buffer (20 µg of LysC/trypsin proteinase (Thermo Fisher, A41007) in 1 ml of digestion buffer (1 mM EDTA, 50 mM Tris-HCl (pH 8) and 0.1 M NaCl)) at 37 °C overnight (16–24 h), as previously reported[25]. Digested gels were washed in 1× PBS two times for 15 min each before proceeding to staining.

For blank gels without embedded biological specimens, 20 µl of initiator solution was added to 411 µl of gelation solution in a 1.5-ml centrifuge tube inside the glove bag. The tube was flipped upside down five times for mixing. The activated gelation solution was added to a constructed gelation chamber (Extended Data Fig. 3e). The gelation chamber was placed in an airtight humidified chamber, taken out of the glove bag and incubated at room temperature (~24 °C) in the dark for 1 h (Extended Data Fig. 4a) or for various durations of time (Extended Data Fig. 4b). Gels were cut into ~0.5 × 0.5 cm shapes and expanded by washes in double-distilled water five times for 5 min each.

## Immunostaining and imaging of expanded cell culture and tissue slices

All of the following steps were performed without shaking, unless otherwise noted. Gels containing brain tissue or cell culture were incubated in blocking solution (0.5% Triton X-100 and 5% normal donkey serum (Jackson ImmunoResearch, 017-000-121) in 1× PBS) for 2 h at room temperature (~24 °C). Gels containing brain tissue or cell culture were then incubated with primary or tertiary antibodies, respectively (see Supplementary Table 3), in staining buffer (0.25% Triton X-100 and 5% normal donkey serum in 1× PBS) at 4 °C overnight (12–24 h). Gels were washed in washing buffer (0.1% Triton X-100 in 1× PBS) four times for 30 min each on a shaker at 40 rpm at room temperature (~24 °C). Gels containing brain tissue were then incubated with secondary antibodies diluted in staining solution at 4 °C overnight (12–24 h) and washed in washing buffer two times for 30 min each on a shaker at 40 rpm at room temperature (~24 °C). Immunostained gels were fully expanded via three to five 20-min washes with 10 ml of double-distilled water in an imaging plate (MatTek, P384G-1.5-10872-C). DAPI staining was performed during the first expansion wash (Thermo Fisher, D1306; dilution of 10 mg ml$^{-1}$ DMSO stock in double-distilled water, 1:1,000, 10 ml).

Gels containing kidney or spleen tissue were incubated in NHS staining solution (Alexa Fluor 488 NHS Ester; Thermo Scientific, A20000; dilution of 10 mg ml$^{-1}$ DMSO stock in 1× PBS, 1:50, 1 ml) at 4 °C overnight (12–24 h) and washed in 1× PBS three times (20 min each) on a shaker at 40 rpm at room temperature (~24 °C). NHS-stained gels were fully expanded via three to five 20-min washes with 10 ml of double-distilled water on an imaging plate (MatTek, P384G-1.5-10872-C).

20ExM-processed sample images were acquired using a Nikon CSU-W1 confocal microscope with a ×4/0.2-NA air objective, a ×10/0.45-NA air objective or a ×40/1.15-NA water-immersion objective, 100% laser power and 300–500 ms exposure time.

The confocal images in Fig. 2a were collapsed to two dimensions using maximum intensity projection, and contrast was adjusted with Fiji's autoscaling function. Confocal images in Fig. 2d,i were adjusted with Fiji's autoscaling function. Confocal images in Fig. 3b were background subtracted using Fiji's rolling ball algorithm with a radius of 50 pixels, collapsed to two dimensions using maximum intensity projection and passed through a two-dimensional Gaussian filter ($\sigma = 1$). The confocal images in Extended Data Fig. 5a,e–h were collapsed to two dimensions using maximum intensity projection, and contrast was adjusted with Fiji's autoscaling function and manually adjusted to improve contrast for the stained structures of interest.

## Expansion factor and resolution measurement

Expansion factors for each sample were determined by imaging whole specimens (tissues and cultured cells) with a confocal microscope before and after the expansion. The expansion factor was determined by measuring the distance between two landmarks in the specimens (Supplementary Table 2)[44]. For samples described in Supplementary Note 2, we also measured the physical gel size with a ruler immediately after gelation and after full expansion.

Resolutions for confocal images in Fig. 2i,j were determined by performing block-wise FRC on a pair of two images that captured the same region with Fiji plugin NanoJ-SQUIRREL's Calculate FRC-Map function[14].

## Peak-to-peak distance measurement

For microtubule analysis, the cross-section line intensity profile was measured over a box area, with the long axis perpendicular to the microtubule and the short axis covering ~185 nm in biological length, using Fiji's line selection tool. The intensity was averaged along the long axis, and the line intensity profile was fitted with a double Gaussian function to detect the two peaks in fluorescence intensity in Python (source code is available at github.com/shiwei-w/20ExM). The distance between the two peaks was measured as the peak-to-peak distance of the microtubule sidewalls.

## r.m.s. error measurement

r.m.s. error measurement was performed similar to as described in previous studies[16]. For *xy* plane analysis, postexpansion confocal images

were passed through a Gaussian filter ($\sigma = 4$), background subtracted using Fiji's rolling ball algorithm with a radius of 50 pixels and collapsed to two dimensions using maximum intensity projection. Pre-expansion images and processed postexpansion confocal images were registered using rigid body registration in Fiji (TurboReg → Scaled Rotation/Accurate/Manual). The images were then nonrigidly registered, and deformation vector fields were calculated in MATLAB (source code is available at github.com/shiwei-w/20ExM).

For analysis in the *xz* or *yz* plane, confocal image *z* stacks of the same brain region were collected and projected onto *xz* and *yz* planes using Fiji's orthogonal view tool and passed through a Gaussian filter ($\sigma = 4$). Both pre- and postexpansion confocal images were registered using rigid body registration and nonrigidly registered in MATLAB in the same fashion as the *xy* plane analysis.

### Autocorrelation and protein enrichment analysis of synaptic nanocolumn

The synaptic nanoarchitecture analysis used in this study was based on previously described methods, specifically autocorrelation ($g_a(r)$) and protein enrichment analysis[10,15,18]. Source code is available at github.com/shiwei-w/20ExM.

For autocorrelation, synapses were identified manually by observing the juxtaposition of presynaptic and postsynaptic clusters[10]. Postexpansion 20ExM images were background subtracted using Fiji's rolling ball algorithm with a radius of 50 pixels, as previously described[10]. The autocorrelation function ($g_a(r)$) in three dimensions measured the likelihood of finding a similar signal at a distance ($r$) from a given signal. This function quantified the heterogeneity of the measured signal within a given volume. To normalize the autocorrelation of each synaptic cluster, the synaptic cluster was compared to an object with the same shape and volume but a homogeneous voxel intensity, which was set to the average intensity of the synaptic cluster. Consequently, a synaptic cluster with uniform intensity would exhibit baseline $g_a(r)$ values at all radii, whereas local intensity peaks within a synaptic cluster would result in higher $g_a(r)$ values over a radius corresponding to the size of the high-intensity region, which then decayed outside of that radius.

For protein enrichment analysis, a cross-enrichment analysis was performed to analyze the distribution of two different protein clusters in relation to each other. This involved measuring the average voxel intensity of one protein cluster (referred to as the 'measured cluster') at various distances from the point of peak intensity in the other protein cluster (referred to as the 'reference cluster', which was shifted in space as previously defined[10]). The measured cluster's intensity values were normalized by comparing them to the average intensity at corresponding distances from the peak intensity point in the reference cluster. To establish this baseline, an object with the same shape and volume as the measured cluster was used, and its voxel intensities were set to the average intensity of the measured cluster. Regions within the measured cluster that exhibited high local intensity would result in normalized intensity values greater than 1.

### Quantification of NPCs

We performed 20ExM with intact NUP96::Neon-AID DLD-1 cells and imaged the NPCs on the top and bottom of the nuclei, tangential to the imaging *z* plane. We manually identified NPCs in seven cells from two culture batches based on the characteristic ring structure with at least four visible corners in top view. To measure the radius of individual NPCs, we used Fiji's radial profile plot plugin to acquire radial intensity distribution and take the peak of the distribution as the radius, as in a previous study[6]. To quantify the number of corners per NPC, we used a previously reported 'Counting Corners' algorithm[20] ($\alpha = 0.93$, threshold = 0.6) that divides each NPC into eight sectors and counts how many sectors contain signals above a given threshold. We then measured the distance between adjacent corners, as determined by the Counting Corners algorithm using Fiji's line selection tool. The line intensity profile was plotted, and the distance between the two peaks was measured as the corner-to-corner distance.

### SNR quantification

We adopted the method for quantifying SNR from a previous study[10] and applied it to the dataset used for synaptic nanocolumn analysis (Fig. 3b). In summary, the images were background subtracted using Fiji's rolling ball algorithm with a radius of 50 pixels. Subsequently, we binarized the image using a threshold calculated as seven times the standard deviation of the average intensity of manually identified background regions, selected every 10–15th slice of the *z* stack. Synapses were identified by selecting the largest three-dimensional connected components[10]. Finally, SNR was determined by dividing the signal intensity by the standard deviation of the background.

### Reporting summary

Further information on research design is available in the Nature Portfolio Reporting Summary linked to this article.

## Data availability

Source and processed imaging data generated in this study are available on Open Science Framework at https://osf.io/kezgs. The source code and data used for synaptic nanocolumn analysis are available on GitHub at https://github.com/shiwei-w/20ExM. Source data are provided with this paper.

## Code availability

The custom code used in this study is available on GitHub at https://github.com/shiwei-w/20ExM.

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

## Acknowledgements

We thank S. D. Brucks, J. W. Alty, E. Badding, S. Y. Lee, D. E. Kim, Q. Yang, J. Kang, M. E. Schroeder, T. B. Tarr, R. Gao, D. Goodwin, K. Titterton, N. Han, B. An, C. Ortiz Cordero, D. Syangtan and D.-A. Mahbuba for thoughtful discussions. We thank T. Schwartz and O. Akkermans at MIT for providing precious materials. We thank the Microscopy Resources on the North Quad (MicRoN) core facility at Harvard Medical School for equipment, support and expertise. Figure 1 was created with Biorender.com. The research reported in this publication was supported by the National Institutes of Health grants R01EB024261, R01MH124606, R01AG087374 and R01AG070831 (E.S.B.), R01AI055258 (L.L.K.) and 1RF1MH124606, 1RF1MH124611 and 1RF1MH128861 (P.Y.). S.W. was supported by the MIT Presidential Graduate Fellowship. T.W.S. and R.B.M. were supported by National Science Foundation Graduate Research Fellowship grants 1122374 and 2140743, respectively. R.B.M. was supported by a National Institutes of Health Molecular Biophysics Training grant (NIGMS T32 GM008313). L.L.K. and E.S.B. acknowledge Open Philanthropy and Good Ventures for funding. E.S.B. acknowledges HHMI, A. Aziz, L. Yang, J. Doerr, T. Stocky, A. Shah, L. McGovern, K. Octavio and the Alana Down Syndrome Center. This project has received funding from the European Research Council under the European Union's Horizon 2020 research and innovation programme (grant agreement 835102; E.S.B.).

## Author contributions

T.W.S. initiated the work with Y.L. and C.Z. for the initial phase of the project. S.W., T.W.S., Y.L., C.Z. and H.B.Y. contributed key ideas. S.W., T.W.S., R.B.M., H.S., P.Y. and H.B.Y. designed and performed experiments to develop and validate the protocol. S.W. developed the final form of 20ExM. T.W.S., S.W., Y.L. and K.S.L. prepared cells. K.S.L. prepared fixed mouse brains. S.W., T.W.S. and H.B.Y. interpreted data and performed data analysis. S.W. and T.W.S. wrote and edited the manuscript. L.L.K. and E.S.B. supervised the project, initiated the work, contributed key ideas, designed experiments, helped with data analysis and interpretation and edited the manuscript.

## Competing interests

S.W., T.W.S., H.B.Y., Y.L., L.L.K. and E.S.B. are co-inventors on a patent application for 20ExM. E.S.B. is cofounder of a company seeking to deploy applications of ExM-related technologies. P.Y. is cofounder, director and consultant of Ultivue, Inc., and Digital Biology, Inc. The other authors declare no competing interests.

## Additional information

**Extended data** is available for this paper at https://doi.org/10.1038/s41592-024-02454-9.

**Correspondence and requests for materials** should be addressed to Laura L. Kiessling or Edward S. Boyden.

**Extended Data Fig. 1 | Molecular mechanism of DMAA gel polymerization.**
DMAA polymerization follows similar initiation and propagation steps as bis-acrylamide gels. However, the crosslinking is achieved by hydrogen extraction and radical transfer (branching). Since the intermediate radicals in this reaction are especially susceptible to reaction with oxygen, the effectiveness of the branching (crosslinking) step will be impacted by the concentration of dissolved oxygen in the gelation solution (ref. 11).

a

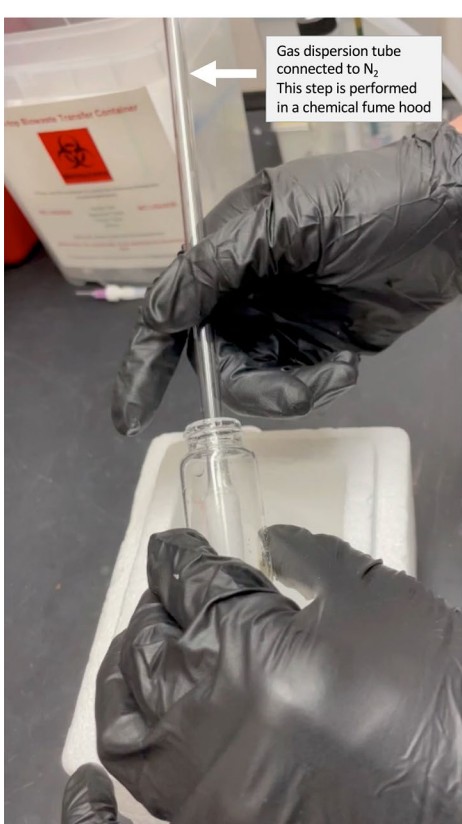

Gas dispersion tube connected to N₂
This step is performed in a chemical fume hood

b

Connected to N₂

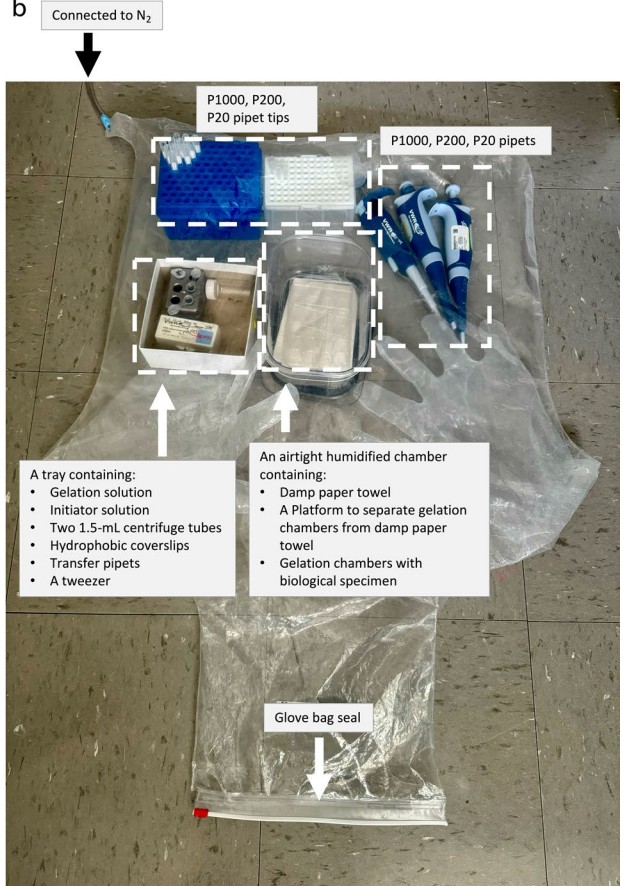

P1000, P200, P20 pipet tips

P1000, P200, P20 pipets

A tray containing:
• Gelation solution
• Initiator solution
• Two 1.5-mL centrifuge tubes
• Hydrophobic coverslips
• Transfer pipets
• A tweezer

An airtight humidified chamber containing:
• Damp paper towel
• A Platform to separate gelation chambers from damp paper towel
• Gelation chambers with biological specimen

Glove bag seal

**Extended Data Fig. 2 | Oxygen-control setup of 20ExM. (a)** The gas dispersion tube is connected to a compressed nitrogen cylinder. With minimal N₂ flow, the gas dispersion tube is placed within the gelation solution, with the sponge part fully wetted and generating bubbles. If the N₂ flow is too strong, the gelation solution will evaporate rapidly and freeze. **(b)** The glove bag is connected to a compressed nitrogen cylinder. All tools required are listed in the figure. All gelation steps are conducted at room temperature (no ice or ice block is needed).

## Gelation chamber construction for tissue

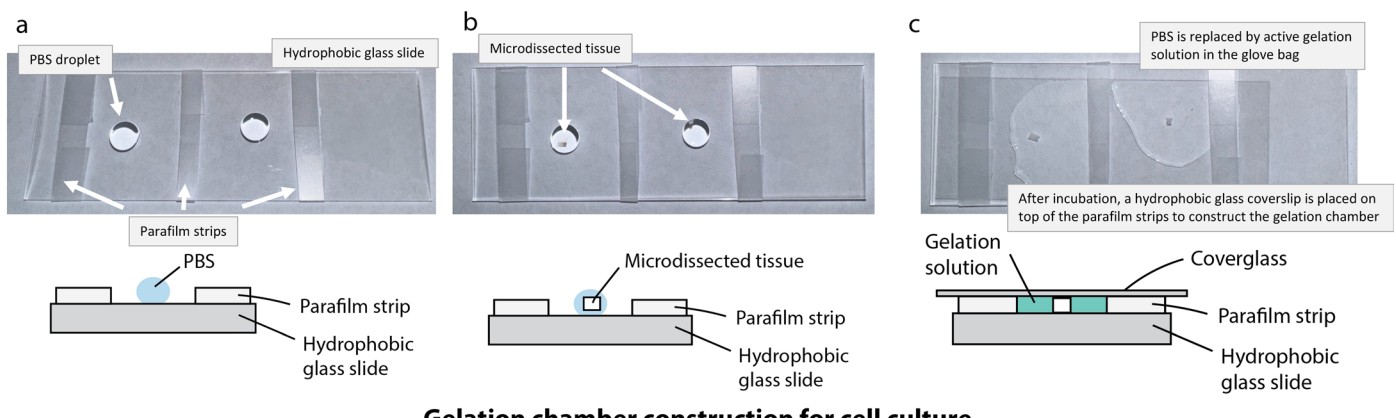

## Gelation chamber construction for cell culture

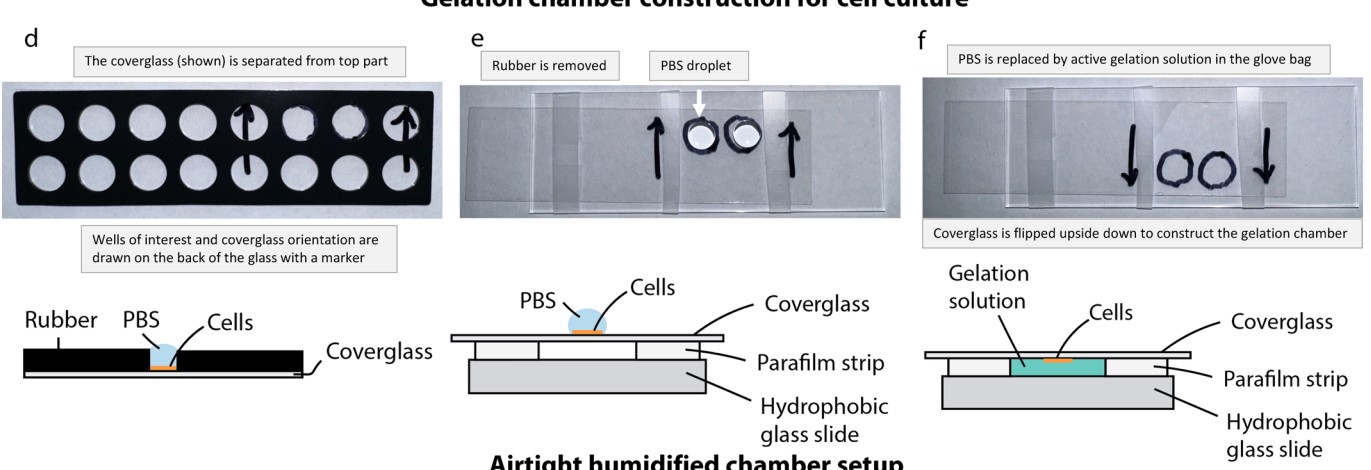

## Airtight humidified chamber setup

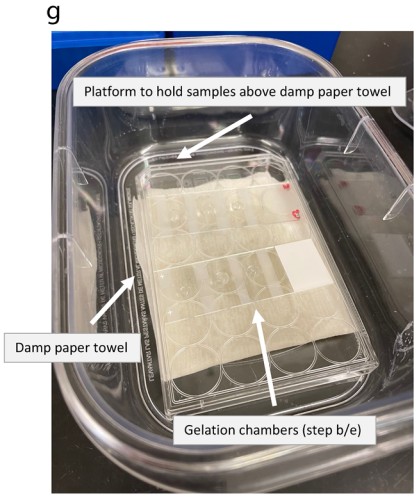

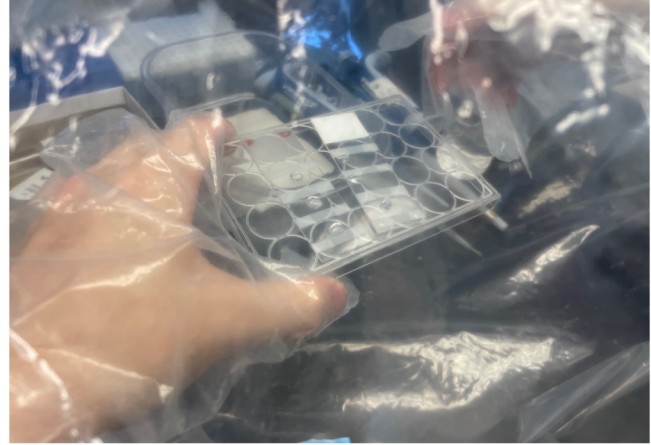

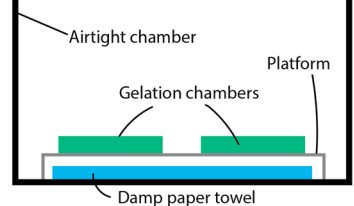

**Extended Data Fig. 3 | Examples of gelation chamber construction and handling. (a-c)** Example gelation chamber construction for tissue. **(d-f)** Example gelation chamber construction for cell culture. The capping of the gelation chamber **(c,f)** is performed within the glove bag. **(g)** Example setup of the airtight humidified chamber, as described in Extended Data Fig. 2b. **(h)** Example of handling glass slides inside the glove bag. The platform (plate lid) can be used to move glass slides inside the glove bag.

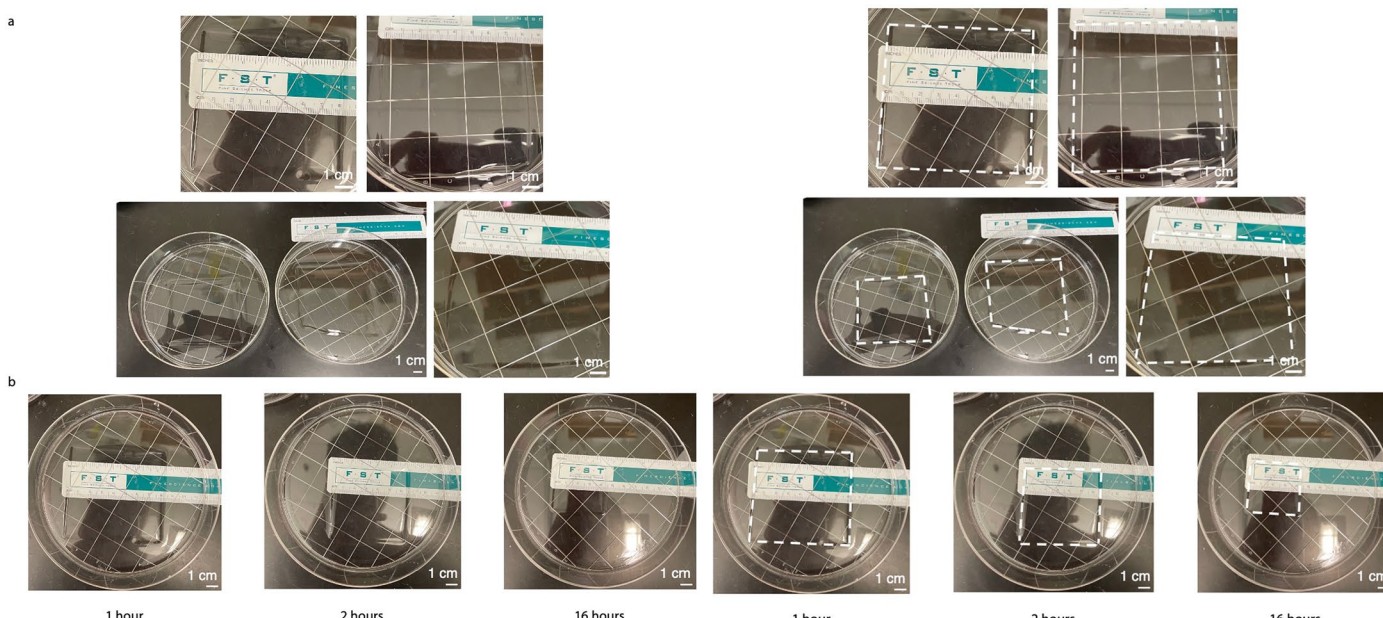

1 hour
16x

2 hours
13x

16 hours
8x

1 hour
16x

2 hours
13x

16 hours
8x

**Extended Data Fig. 4 | Reproducibility and time dependence of 20ExM without biological specimen embedded. (a)** Fully expanded gels made in 5 different batches with gelation time of 1 hour, all reaching an expansion factor of ~16x. Unexpanded gels were cut into ~0.5 × 0.5 cm shapes (note, the shapes were not exactly rectangular – they were the shapes shown) and expanded gels are ~8 × 8 cm. **(b)** Fully expanded gels with various gelation times. Unexpanded gels were cut into ~0.5 × 0.5 cm shapes (note, the shapes were not exactly rectangular – they were the shapes shown). Expansion factor and gelation time are indicated in the figure. Right side: same images as the left side, with border of gels highlighted by white dotted lines. Scale bars: 1 cm.

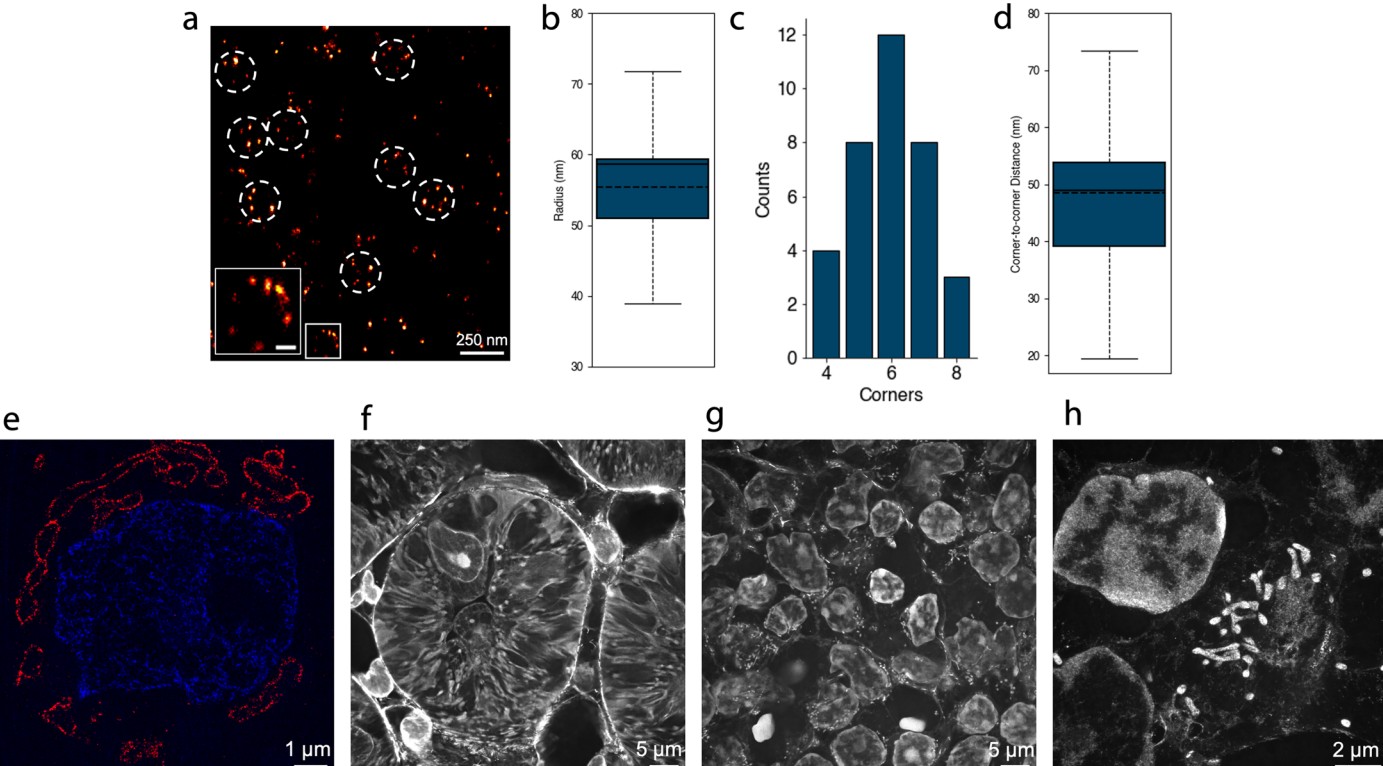

**Extended Data Fig. 5 | Applications of 20ExM. (a)** Confocal image (maximum intensity projection; from one representative experiment from two culture batches) of expanded *NUP96::Neon-AID* DLD-1 cells with pre-expansion anti-mNeonGreen staining, with some NPCs highlighted by white dotted circles. Inset: magnified view of the white boxed region. Note: We noticed some single-puncta signals that did not participate in a ring. These could be parts of other nuclear pore complexes (for example, partially assembled), or incompletely stained nuclear pore complexes, or nonspecific staining. We did not use a special NPC preparation strategy, as is common for microtubules. More specialized fixation and extraction methods, such as permeabilization with detergent prior to fixation, cryofixation with methanol, or extracting nuclei from intact cells prior to staining and expansion, might in principle further improve staining quality (of course, such practices, while they may improve the appearance of NPCs, do not resemble a typical ExM user's application, nor is it representative of methods that optimally preserve general biological ultrastructure). However, in earlier best-practices ExM studies visualizing nuclear pore complexes, even with specialized fixation, purification, and staining methods designed to optimize nuclear pore appearance, the investigators often observed single puncta that did not appear to be part of a ring (ref. 20). This reference, which claimed similar resolution to what we show here (albeit with an iterative form of expansion microscopy), also reported images and numbers similar to ours regarding the shapes of nuclear pores, the number of corners of each nuclear pore, and the diameters of nuclear pores. Our goal in the current study was not to study NPCs, but rather to validate the resolution and gel-contributed error of 20ExM with NPCs. Furthermore, our goal was not to better earlier iterative methods like iExM or ExR, but rather to show that such performance could be achieved in a single step. Since our current protocol was sufficient to for these purposes, and indeed, matched the performance of previous best-practices expansion microscopy protocols when tested against nuclear pore visualization, we did not pursue further optimization. Expansion factor: 22.8 ± 0.4 as measured by physical gel size (n = 2 culture batches). Brightness and contrast settings: first set by Fiji's auto-scaling function and then manually adjusted to improve contrast for the stained structures of interest; quantitative analysis in b–d was conducted on raw image data. **(b)** Box plot of radius of 35 NPCs in top view (from n = 7 cells from two culture batches; black horizontal line, median; dotted horizontal line, mean; upper edge of the box, first quartile; lower edge of the box, third quartile; top dotted line extended from the box represents first quartile minus 1.5x the inter-quartile range; bottom dotted line extended from the box represents third quartile plus 1.5x the inter-quartile range). **(c)** Population data for 35 NPCs (from n = 7 cells from two culture batches), showing a histogram of corners per NPCs. **(d)** Box plot of distances between adjacent corners of 35 NPCs in top view (from n = 108 measurements of 35 NPCs from 7 cells from two culture batches; black horizontal line, median; dotted horizontal line, mean; upper edge of the box, first quartile; lower edge of the box, third quartile; top dotted line extended from the box represents first quartile minus 1.5x the inter-quartile range; bottom dotted line extended from the box represents third quartile plus 1.5x the inter-quartile range). **(e)** Confocal image (maximum intensity projection; from one representative experiment from two culture batches) of expanded HEK293 cells with pre-expansion anti-TOM20 (red) staining and post-expansion DAPI (blue) staining. **(f)** Confocal image (single-xy plane; from one representative experiment of two kidney slides from one mouse) of mouse kidney after performing 20ExM and post-expansion NHS-AlexaFluor488 staining. **(g)** Confocal image (single-xy plane; from one representative experiment of two spleen slides from one mouse) of mouse spleen after performing 20ExM and post-expansion NHS-AlexaFluor488 staining. **(h)** Confocal image (single-xy plane; from one representative experiment of two spleen slides from one mouse) of mouse spleen after performing 20ExM and post-expansion NHS-AlexaFluor488 staining. Expansion factor for f–h: 16.5 ± 0.4 (n = 2 spleen, 2 kidney sections from 1 mouse; measured by physical gel size). Scale bars are provided in biological units (that is, physical size divided by expansion factor) throughout all figures: **(a)** 250 nm and 50 nm in inset, **(e)** 1 μm, **(f)** 5 μm, **(g)** 5 μm, **(h)** 2 μm.

# Reporting Summary

## Statistics

For all statistical analyses, confirm that the following items are present in the figure legend, table legend, main text, or Methods section.

| n/a | Confirmed | |
|---|---|---|
| ☐ | ☒ | The exact sample size (*n*) for each experimental group/condition, given as a discrete number and unit of measurement |
| ☐ | ☒ | A statement on whether measurements were taken from distinct samples or whether the same sample was measured repeatedly |
| ☒ | ☐ | The statistical test(s) used AND whether they are one- or two-sided<br>*Only common tests should be described solely by name; describe more complex techniques in the Methods section.* |
| ☒ | ☐ | A description of all covariates tested |
| ☒ | ☐ | A description of any assumptions or corrections, such as tests of normality and adjustment for multiple comparisons |
| ☐ | ☒ | A full description of the statistical parameters including central tendency (e.g. means) or other basic estimates (e.g. regression coefficient) AND variation (e.g. standard deviation) or associated estimates of uncertainty (e.g. confidence intervals) |
| ☒ | ☐ | For null hypothesis testing, the test statistic (e.g. *F*, *t*, *r*) with confidence intervals, effect sizes, degrees of freedom and *P* value noted<br>*Give P values as exact values whenever suitable.* |
| ☒ | ☐ | For Bayesian analysis, information on the choice of priors and Markov chain Monte Carlo settings |
| ☒ | ☐ | For hierarchical and complex designs, identification of the appropriate level for tests and full reporting of outcomes |
| ☒ | ☐ | Estimates of effect sizes (e.g. Cohen's *d*, Pearson's *r*), indicating how they were calculated |

*Our web collection on statistics for biologists contains articles on many of the points above.*

## Software and code

Policy information about availability of computer code

| | |
|---|---|
| Data collection | Spinning disk confocal microscope with NIS-Element Advanced Research |
| Data analysis | Python (version 3.11.7), Fiji (version 2.14.0/1.54f; Built-in: Rolling Ball Function, Gaussian Filter Function, Auto-scaling Function, Line Selection Tool; Plug-in: Radial Profile 1.0, NanoJ-SQUIRREL 1.0), Microsoft Excel (version 16.87), MATLAB (version R2020a). Custom codes used for Distortion Analysis, Peak-to-peak Distance Analysis, and Autocorrelation and Enrichment Analysis are available on GitHub at https://github.com/shiwei-w/20ExM. |

For manuscripts utilizing custom algorithms or software that are central to the research but not yet described in published literature, software must be made available to editors and reviewers. We strongly encourage code deposition in a community repository (e.g. GitHub). See the Nature Portfolio guidelines for submitting code & software for further information.

## Data

Policy information about availability of data

All manuscripts must include a data availability statement. This statement should provide the following information, where applicable:
- Accession codes, unique identifiers, or web links for publicly available datasets
- A description of any restrictions on data availability
- For clinical datasets or third party data, please ensure that the statement adheres to our policy

Source and processed imaging data generated in this study is available on Open Science Framework at https://osf.io/kezgs. The source data used for synaptic

nanocolumn analysis is available on GitHub at https://github.com/shiwei-w/20ExM.

## Human research participants

Policy information about studies involving human research participants and Sex and Gender in Research.

| | |
|---|---|
| Reporting on sex and gender | N/A |
| Population characteristics | N/A |
| Recruitment | N/A |
| Ethics oversight | N/A |

Note that full information on the approval of the study protocol must also be provided in the manuscript.

# Field-specific reporting

Please select the one below that is the best fit for your research. If you are not sure, read the appropriate sections before making your selection.

☒ Life sciences          ☐ Behavioural & social sciences          ☐ Ecological, evolutionary & environmental sciences

For a reference copy of the document with all sections, see nature.com/documents/nr-reporting-summary-flat.pdf

# Life sciences study design

All studies must disclose on these points even when the disclosure is negative.

| | |
|---|---|
| Sample size | We have clearly described the number of samples we tried for each experiments and results. All of the conclusions were based on n>1 replications. As we are demonstrating a new technology, we performed two to three biological replicates with the same protocol to demonstrate the reproducibility of the protocol. |
| Data exclusions | None |
| Replication | For all experiments, we performed two to three independent biological replicates. All the replications were successful. |
| Randomization | Since we are just demonstrating our technology, instead of, for example, determining the differences of treatment and control groups, randomization was not relevant to this study. |
| Blinding | Since we are just demonstrating our technology, instead of, for example, determining the differences of treatment and control groups, blinding was not relevant to this study. |

# Reporting for specific materials, systems and methods

We require information from authors about some types of materials, experimental systems and methods used in many studies. Here, indicate whether each material, system or method listed is relevant to your study. If you are not sure if a list item applies to your research, read the appropriate section before selecting a response.

## Materials & experimental systems

| n/a | Involved in the study |
|---|---|
| ☐ | ☒ Antibodies |
| ☐ | ☒ Eukaryotic cell lines |
| ☒ | ☐ Palaeontology and archaeology |
| ☐ | ☒ Animals and other organisms |
| ☒ | ☐ Clinical data |
| ☒ | ☐ Dual use research of concern |

## Methods

| n/a | Involved in the study |
|---|---|
| ☒ | ☐ ChIP-seq |
| ☒ | ☐ Flow cytometry |
| ☒ | ☐ MRI-based neuroimaging |

## Antibodies

| | |
|---|---|
| Antibodies used | anti-Beta tubulin (Rabbit, Abcam, ab6046), |

| Antibodies used | anti-mNeonGreen (Mouse, Proteintech, 32f6),<br>anti-TOM20 (Rabbit, Proteintech, 11802-1-AP),<br>anti-rabbit Alexa Fluor 546 (Goat, ThermoFisher, A11035)<br>anti-mouse Alexa Fluor 568 (Goat, ThermoFisher, A21043)<br>anti-goat Alexa Fluor Plus 555 (Donkey, ThermoFisher, A32816)<br>anti-RIM1/2 (Guinea pig, Synaptic Systems, 140205)<br>anti-PSD95 (Mouse, ThermoFisher, MA1-046)<br>anti-GFP (Rabbit, ThermoFisher, A11122)<br>anti-Guinea pig Alexa Fluor 555 (Goat, ThermoFisher, A21435)<br>anti-Mouse Alexa Fluor Plus 647 (Donkey, ThermoFisher, A32787)<br>anti-Rabbit Alexa Fluor Plus 488 (Goat, ThermoFisher, A11008) |
|---|---|
| Validation | All of the antibodies are commercially available. The following validations are performed by the vendors.<br>anti-Beta tubulin (Rabbit, Abcam, ab6046): Validated for Human IHC-P, ICC/IF, IP, and WB<br>anti-mNeonGreen (Mouse, Proteintech, 32f6): Validated for Mouse ICC/IF and ELISA<br>anti-TOM20 (Rabbit, Proteintech, 11802-1-AP): Validated for Human WB, IP, and ICC/IF, and Mouse WB, IHC<br>anti-RIM1/2 (Guinea pig, Synaptic Systems, 140205): Validated for Rat WB and ICC/IF, and Mouse ExM IHC<br>anti-PSD95 (Mouse, ThermoFisher, MA1-046): Validated for Mouse WB, Rat WB and ICC/IF, Human ICC/IF<br>anti-GFP (Rabbit, ThermoFisher, A11122): Validated for Human WB and ICC/IF |

# Eukaryotic cell lines

Policy information about cell lines and Sex and Gender in Research

| Cell line source(s) | HEK 293 from ThermoFisher, catalog no. R70007; Human colorectal adenocarcinoma cells (DLD-1) with homozygous insertion at Nup96 loci containing NeonGreen moiety and an auxin-inducible degron (Nup96::Neon-AID) from laboratory of T. Schwartz and M. Dasso (Ref. 21: Regmi et al., 2020; Ref. 22: Schuller et al., 2021). |
|---|---|
| Authentication | HEK 293 are available at ThermoFisher, catalog no. R70007, Authentication method not specified by vendor;<br>Creation and authentication of the Nup96::Neon-AID DLD-1 cell line was performed in a separate study (Ref. 21: Regmi et al., 2020), Authenticated by PCR assays with insertion-specific primers, imaging assay, and Auxin activation assay. |
| Mycoplasma contamination | The cell lines were not tested for mycoplasma contamination. |
| Commonly misidentified lines<br>(See ICLAC register) | No commonly misidentified lines were used in the study. |

# Animals and other research organisms

Policy information about studies involving animals; ARRIVE guidelines recommended for reporting animal research, and Sex and Gender in Research

| Laboratory animals | Thy1-YFP-H, 6–8 weeks of age. |
|---|---|
| Wild animals | No wild animals were used in the study. |
| Reporting on sex | Used without regard to sex. |
| Field-collected samples | No field-collected samples were used in the study. |
| Ethics oversight | All procedures involving mice (Thy1-YFP-H, 6–8 weeks of age from JAX, used without regard to sex) were performed in accordance with the US National Institutes of Health Guide for the Care and Use of Laboratory Animals and approved by the MIT Committee on Animal Care. |

Note that full information on the approval of the study protocol must also be provided in the manuscript.

