## [Peer Review File · Nature Methods]

Single-shot 20-fold Expansion Microscopy

Corresponding Author: Professor Edward Boyden

Version 0:

Decision Letter:

12th Oct 2023

Dear Ed,

Your Article, "Single-shot 20-fold Expansion Microscopy Enables 18-nm Resolution Imaging on Conventional Microscopes", has now been seen by three reviewers. As you will see from their comments below, although the reviewers find your work of considerable potential interest, they have raised a number of concerns. We are interested in the possibility of publishing your paper in Nature Methods, but would like to consider your response to these concerns before we reach a final decision on publication.

We therefore invite you to revise your manuscript to address these concerns. We ask you to focus on (1) demonstrating on diverse sample types/structures, (2) showing the benefits over 10X expansion, and (3) better quantifying expansion isotropy.

Link Redacted

We hope to receive your revised paper within XX weeks [****ED TO CUSTOMIZE AS NEEDED****]. If you cannot send it within this time, please let us know. In this event, we will still be happy to reconsider your paper at a later date so long as nothing similar has been accepted for publication at Nature Methods or published elsewhere.

OPEN SCIENCE REQUIREMENTS

REPORTING SUMMARY AND EDITORIAL POLICY CHECKLISTS

OK TO DELETE SECTION IF NO GELS OR BLOTS

IMAGE INTEGRITY

DATA AVAILABILITY

All novel DNA and RNA sequencing data, protein sequences, genetic polymorphisms, linked genotype and phenotype data, gene expression data, macromolecular structures, and proteomics data must be deposited in a publicly accessible database, and accession codes and associated hyperlinks must be provided in the "Data Availability" section.

MATERIALS AVAILABILITY

SUPPLEMENTARY PROTOCOL

To help facilitate reproducibility and uptake of your method, we ask you to prepare a step-by-step Supplementary Protocol for the method described in this paper. We [encourage authors to share their step-by-step experimental protocols](https://www.nature.com/nature-research/editorial-policies/reporting-standards#protocols) on a protocol sharing platform of their choice and report the protocol DOI in the reference list. Nature Portfolio's Protocol Exchange is a free-to-use and open resource for protocols; protocols deposited in Protocol Exchange are citable and can be linked from the published article. More details can be found at www.nature.com/protocolexchange/about.

ORCID

Sincerely,
Rita

Rita Strack, Ph.D.
Senior Editor
Nature Methods

Reviewers' Comments:

Reviewer #1:

Remarks to the Author:

Super-resolution microscopy stands as a pivotal tool in cellular biology, with a persistent need for advancement to visualize previously unseen structures. The authors have introduced a new, optimized ExM method capable of a one-step 20-fold expansion. While ExM is already recognized as a cost-effective and robust super-resolution technique, the prevalent method offers only a 4-fold expansion. Achieving expansions greater than 5-fold typically requires multiple expansion steps or specialized equipment. This study introduces a one-step method that allows for a 20-fold expansion without any specialized equipment. In theory, the 20-fold expansion can attain an approximate lateral resolution of 12 nm when used with a standard confocal microscope, given the typical 240 nm lateral resolution of confocal microscopy. Supporting this, the authors noted an accuracy of roughly 17 nm in lateral measurements. I believe this 20ExM method can serve as a valuable tool, offering enhanced "native" resolution without the need for post-imaging processing across various research domains. I recommended this work for publication in the leading journal, Nature Methods, after addressing following concerns.

Major concerns

1. The authors demonstrated that gels with varying gelation times displayed different expansion rates (see Extended Figure 4). It's puzzling to note that none of the time points reached a 20x expansion, while gels containing biological specimens (tissue cells) achieved a 21-fold expansion. It would be beneficial if the authors could provide gel size measurements for both cells and tissues that align with the expansion rates observed in biological samples. Additionally, an insight into how gelation time influences these measurements with biological specimens and their reproducibility would be valuable.
2. Related to 1, authors should provide expansion rates in z-axis of gels and biological specimen.
3. An advantage of 20xExM is its rapid gel expansion, in contrast to the 10xExM gels which necessitate a lengthier expansion process involving several water exchanges. We observed that 10xExM gels begin to contract after reaching a 10x expansion, typically after a few hours. It would be valuable if the authors could provide information regarding the stability of the expanded gels and suggest an optimal timeline for imaging to ensure consistent expansion.
4. The authors utilized the microtubule as a cellular ruler, a fitting choice given the well-studied nature of microtubule structures. However, the measured thickness of microtubules labeled with primary and secondary antibodies, as captured by super-resolution microscopy, ranged from 35-70 nm. This raises questions about the actual sizes of antibody-labeled microtubules. While it's not imperative for this study to validate the true thickness of the microtubules, the authors should offer both expansion factor measurements and error estimates with multiple rulers. To enhance reliability, they might consider using another well-characterized cellular ruler, such as the centrosome.

5. Figures I and J. FRC measurements are influenced by the signal-to-noise (S/N) ratio. It would be beneficial for the authors to include intensity data alongside these results. While pinpointing the precise maximum resolution isn't crucial, it would be enlightening to see structures that approach this estimated maximum resolution. If 20ExM yields a 17 nm lateral resolution, actin might serve as a more appropriate ruler compared to microtubules for 20ExM.
6. The microtubule images appear faint, likely due to the 20-fold enlargement of the stained samples. It would be valuable for the authors to describe a method for achieving high signal-to-noise (S/N) staining and also to provide quantitative measurements of the signal reduction.
7. 8. Similarly to point 7, the RIM1/2 and PSD95 staining images in 20ExM appear faint due to noticeable noise, even though the regions selected by the authors seem clear. This raises concerns about potential bias in measurements based on the regions chosen for quantification. The authors should offer quantitative data on signal intensity and discuss any challenges therein. Moreover, the images in Fig. 3b appear to have excessive contrast, making them look oversaturated.
8. A concern in 20-fold expansion is isotropy in Z-axis. If authors can validate the isotropic expansion in Z-axis using biological specimens, it would be valuable.

Minor concerns

9. DMAA gels are notably softer than the MBA-based 4x ExM gels, and likely requires training in handling. It would be beneficial if the authors could share recommendations and best practices for handling and imaging these gels.

Reviewer #2:

Remarks to the Author:

In this manuscript, Wang et. al. have developed a single-shot expansion microscopy approach with a 20X expansion factor, and a working resolution of 18 nm. This end was achieved through optimizing the ratio of hydrogel components used for 10X expansion, N,N-dimethylacrylamide (DMAA) and sodium acrylate. The authors further manipulated gelation conditions by deoxygenating the gelation solution with nitrogen gas and working in an oxygen free environment. By tuning the potassium persulfate initiator concentration, as well as the gelation time, the authors were able to show convincing microtubule expansion in cell culture, as well neuron synapse architecture of mouse brain tissue.

A one-shot 20X ExM protocol represents a significant advance in expansion microscopy, completely avoiding the cumbersome protocols of iterative ExM, while reaching new ceilings in expansion factor. The additional ability to perform post-expansion labeling is an asset. This is especially highlighted by the visualization of the microtubule lumen, a remarkable feat in super resolution imaging. 20ExM seems relatively straightforward to perform and has the potential to become among the most utilized protocols for large-expansion factor imaging. If the following concerns are addressed, this work merits publication in Nature Methods.

Major Concerns

Validation of 20ExM on more than one organelle and tissue type. In the studies often cited, and previously published in some cases, by the authors in this manuscript (TrEX, 10X, Magnify, Expansion Revealing) multiple cell compartments, and with diverse antibody stains, were described to show proof of concept. The microtubule staining is remarkable, but it is important to see more examples of the protocol in action, on more diverse organelles, especially an example of a membrane-bound organelle.

Extended Figures 2,3, and 4 are not very intuitive. The methods and the text clearly describe how to perform the protocol. However, the extended figures made to visualize the experimental setup are lacking clarity. In Extended Figure 2, a better example of the deoxygenated glove-bag environment would be useful. In Extended Figure 3, it was unclear what divided panels a,b and c from panels d,e, and f. The illustrated schemes make the addition of the gelation solution more understandable, but the figure could benefit from clearer images. It is difficult to gauge the expansion factor in Extended Figure 4 without scale bars and being unable to read the ruler. The scaling of the individual gels also seems different.

Minor Concerns / Questions

An important aspect of this protocol, as well as all expansion protocols, are the ease and feasibility of setting it up in the laboratory. There is a non-trivial difference between the 4 uL of initiator being added to the brain slice, and 20 uL being added to cultured cell. This is true as well for the 16-20 hours of gelation time for brain slices, and 2 hours for cultured cells. Have the authors further tested sections of different tissues to see if the initiator concentration or gelation time will need further adjustments, or do they expect that the 4 uL /20-hour 20 uL / 4-hour benchmarks will serve a diverse spectrum of tissues, or do you expect modifications of these ratios to ensure maximum expansion?

Some of the references to the Extended Figures are incorrect in the Supplementary Note. For example, Line 49 refers to sealing the connection between the tube and the glove bag with tape, which is Extended Figure 2b, and lines 90-91 incorrectly attributes adding nitrogen to the gelation solution to Extended Figure 4a.

Reviewer #3:

Remarks to the Author:

The manuscript by Wang et al describes a novel polymer formulation for performing expansion microscopy. The authors provide a hypothesis that explains why atmospheric oxygen may prevent uniform polymerisation and, hence, uniform expansion of the sample using this polymer. They show that by using an inert atmosphere (in a disposable glove bag) the polymerisation reaction can be carried out robustly and that this combination of methodological development, along with the novel formulation enable unprecedented expansion factors from a single polymerisation reaction.

The manuscript is technically sound and sufficient information is provided to reproduce the work. Indeed, the Supplementary photos and descriptions are extremely useful in guiding the practical implementation of the method, a feature which is often overlooked.

In its current form, the manuscript does not show meaningful application to a biological question and doesn't, therefore, demonstrate any significant advantage over existing approaches for super-resolution imaging. For example, are the images significantly improved (by some metric) compared with iExM, what is the need for 20x over 10x expansion factor?

A significant challenge with expansion microscopy is final image quality and particularly as seen for continuous features such as microtubules, where labelling can be punctate and patchy. It would be useful to see more on this in the manuscript- naively, one might expect a single polymerisation reaction to lead to some improvement in labelling/image quality (less fluorophore/antigen degradation).

A general comment about the narrative around absolute distance measurements in the manuscript. The authors quote a standard deviation in the expansion factor of approximately 8% (21.5 +/- 1.7x). As I understand, in the manuscript, expansion factor is calculated for each gel, with pre- and post-expansion imaging. I think it's unlikely that the majority of users will consider using pre-expansion imaging in this way, and indeed, may lack the expertise to perform the registration of pre-/post-expansion images and accurately derive the expansion factor for a given gel. It should be clear from the manuscript that such an approach is necessary to perform quantitative measurement in the samples.

Version 1:

Decision Letter:

Our ref: NMETH-A53649A

26th Jun 2024

Dear Ed,

Thank you for submitting your revised manuscript "Single-shot 20-fold Expansion Microscopy Enables 18-nm Resolution Imaging on Conventional Microscopes" (NMETH-A53649A). It has now been seen by the original referees and their comments are below. The reviewers find that the paper has improved in revision, and therefore we'll be happy in principle to publish it in Nature Methods, pending minor revisions to satisfy the referees' final requests and to comply with our editorial and formatting guidelines.

We sent your responses to the remaining concerns to the other reviewers. The feedback we received was that the concerns were valid, but that your responses adequately addressed them. We ask that you provide a full point-by-point rebuttal upon resubmission and that you update the manuscript to include the additional comparisons you did for the rebuttal and discuss all the raised concerns.

TRANSPARENT PEER REVIEW

ORCID

Sincerely,
Rita

Rita Strack, Ph.D.
Senior Editor
Nature Methods

Reviewer #1 (Remarks to the Author):

Expansion Microscopy (ExM) has become an essential toolkit for broad applications in cell and developmental biology, offering super-resolution capabilities to researchers without the need for expensive super-resolution microscopes. Current ExM techniques are limited to 4-10-fold expansion, with greater expansions typically requiring iterative processes that are time-consuming and complex. This new 20ExM method improves the chemistry to enable a one-step process, achieving a 20-fold expansion. Although the use of ExM is significantly increasing worldwide, a major concern and limitation of these methods is the preservation of biological architectures due to the physical expansion and protein digestion processes. Particularly, the 20-fold expansion raises serious concerns, and strong evidence must be provided to ensure the preservation of biological structures. The significant impact of the 20ExM method lies in its one-step expansion process, which is expected to minimize potential damage to biological architectures compared to multi-step expansions.

In the revised manuscript, the authors have addressed some of the major concerns raised in the initial submission, such as isotropic expansion and gel stability, by performing new experiments. However, the new images of nuclear pores and existing microtubules were not convincing in addressing these concerns. Instead, these images suggest that 20ExM may cause serious distortions in biological architectures, limiting its applications. The mouse brain images at low magnifications are beautiful, indicating that large-scale structures are likely well preserved. However, the primary users of 20ExM will be studying nanometer-scale structures, not just ~100 nm structures, which can already be studied with the original or existing ExM methods. Authors must provide strong evidence that nano-scale structures are preserved using 20ExM before publication in a leading methods journal, Nature Methods.

Major concern

I understand and appreciate the effort to visualize nuclear pores as an additional nanoscale cellular ruler alongside microtubules. However, I have some concerns regarding the quality of the nuclear pore images. They appear unconvincing, making it difficult for readers to accept the quantifications. Many strong Nup96 puncta outside the circle marks are randomly distributed, and the nuclear pores marked as circles by authors do not resemble known nuclear pores. These images, along with some microtubule images, raise concerns about the preservation of nanoscale biological structures using 20ExM, potentially due to the 20-fold physical expansion. Additionally, in the new Supplementary Figure 3, the authors demonstrated isotropic expansion in the Z-axis and calculated approximately 5% distortion. However, the images appear significantly different before and after ExM, making it challenging to agree with the 5% distortion claim. There are many new signals in the post-20ExM images, and shapes differ significantly, even in regions where both images show signals. Again, these results raise concerns that 20ExM induces significant distortion in axial structures. The new mitochondria and brain images, while demonstrating micron/submicron scale preservation, do not convincingly prove the preservation of nanoscale architecture, which is a primary focus for most 20ExM users. Artificial distortion due to physical expansion and/or protein digestion remains a significant concern in ExM, especially with expansions larger than the original 4-fold. The authors must provide validated structures that ensure nanoscale structural preservation, which is the principal concept of 20ExM for a broad audience. This might be achievable using centrosome 9-cartwheel structures, among others. If antibody size is a concern, labeling can be performed by expressing tag-conjugated proteins such as GFP (enhanced by commercial nanobodies for GFP to minimize extra size), SNAP, etc.

Minor concern

The authors addressed the expected signal reductions due to 20ExM in a step-by-step response. However, this significant limitation could severely restrict its applications, as it will not work with typical primary-secondary antibody staining. While this limitation does not diminish the impact of the technology, the authors must clearly and extensively discuss these points in the Main text. Specifically, they should address the expected signal reductions, the necessity for tertiary antibody staining, the potential limitations, and possible ways to overcome these technical barriers. These aspects are crucial for readers and future users.

Version 2:

Decision Letter:

9th Sep 2024

Dear Ed,

I am pleased to inform you that your Article, "Single-shot 20-fold Expansion Microscopy", has now been accepted for publication in *Nature Methods*. The received and accepted dates will be August 26, 2023 and Sep 9, 2024. This note is intended to let you know what to expect from us over the next month or so, and to let you know where to address any further questions.

Over the next few weeks, your paper will be copyedited to ensure that it conforms to *Nature Methods* style. Once your paper is typeset, you will receive an email with a link to choose the appropriate publishing options for your paper and our Author Services team will be in touch regarding any additional information that may be required. It is extremely important that you let us know now whether you will be difficult to contact over the next month. If this is the case, we ask that you send us the contact information (email, phone and fax) of someone who will be able to check the proofs and deal with any last-minute problems.

Please note that *Nature Methods* is a Transformative Journal (TJ). Authors may publish their research with us through the traditional subscription access route or make their paper immediately open access through payment of an article-processing charge (APC). Authors will not be required to make a final decision about access to their article until it has been accepted. [Find out more about Transformative Journals](https://www.springernature.com/gp/open-research/transformative-journals)

If you are active on Twitter/X, please e-mail me your and your coauthors' handles so that we may tag you when the paper is published.

You can now use a single sign-on for all your accounts, view the status of all your manuscript submissions and reviews, access usage statistics for your published articles and download a record of your refereeing activity for the *Nature* journals.

Please note that you and any of your coauthors will be able to order reprints and single copies of the issue containing your article through *Nature Portfolio's* reprint website, which is located at <http://www.nature.com/reprints/author-reprints.html>. If there are any questions about reprints please send an email to author-reprints@nature.com and someone will assist you.

Best regards,
Rita

Rita Strack, Ph.D.
Senior Editor
Nature Methods

Visit the Springer Nature Editorial and Publishing website at http://editorial-jobs.springernature.com?utm_source=ejP_NMeth_email&utm_medium=ejP_NMeth_email&utm_campaign=ejp_Nmeth > www.springernature.com/editorial-and-publishing-jobs for more information about our career opportunities. If you have any questions please click [here](mailto:editorial.publishing.jobs@springernature.com) .*

Open Access This Peer Review File is licensed under a Creative Commons Attribution 4.0 International License, which permits use, sharing, adaptation, distribution and reproduction in any medium or format, as long as you give appropriate credit to the original author(s) and the source, provide a link to the Creative Commons license, and indicate if changes were made. In cases where reviewers are anonymous, credit should be given to 'Anonymous Referee' and the source.

Reviews

Reviewer #1:

Remarks to the Author:

Expansion Microscopy (ExM) has become an essential toolkit for broad applications in cell and developmental biology, offering super-resolution capabilities to researchers without the need for expensive super-resolution microscopes. Current ExM techniques are limited to 4-10-fold expansion, with greater expansions typically requiring iterative processes that are time-consuming and complex. This new 20ExM method improves the chemistry to enable a one-step process, achieving a 20-fold expansion.

Although the use of ExM is significantly increasing worldwide, a major concern and limitation of these methods is the preservation of biological architectures due to the physical expansion and protein digestion processes. Particularly, the 20-fold expansion raises serious concerns, and strong evidence must be provided to ensure the preservation of biological structures. The significant impact of the 20ExM method lies in its one-step expansion process, which is expected to minimize potential damage to biological architectures compared to multi-step expansions.

In the revised manuscript, the authors have addressed some of the major concerns raised in the initial submission, such as isotropic expansion and gel stability, by performing new experiments. However, the new images of nuclear pores and existing microtubules were not convincing in addressing these concerns. Instead, these images suggest that 20ExM may cause serious distortions in biological architectures, limiting its applications. The mouse brain images at low magnifications are beautiful, indicating that large-scale structures are likely well preserved. However, the primary users of 20ExM will be studying nanometer-scale structures, not just ~100 nm structures, which can already be studied with the original or existing ExM methods. Authors must provide strong evidence that nano-scale structures are preserved using 20ExM before publication in a leading methods journal, Nature Methods.

Major concern

I understand and appreciate the effort to visualize nuclear pores as an additional nanoscale cellular ruler alongside microtubules. However, I have some concerns regarding the quality of the nuclear pore images. They appear unconvincing, making it difficult for readers to accept the quantifications. Many strong Nup96 puncta outside the circle marks are randomly distributed, and the nuclear pores marked as circles by authors do not resemble known nuclear pores. These images, along with some microtubule images, raise concerns about the preservation of nanoscale biological structures using 20ExM, potentially due to the 20-fold physical expansion. Additionally, in the new Supplementary Figure 3, the authors demonstrated isotropic expansion in the Z-axis and calculated approximately 5% distortion. However, the images appear significantly different before and after ExM, making it challenging to agree with the 5% distortion

claim. There are many new signals in the post-20ExM images, and shapes differ significantly, even in regions where both images show signals. Again, these results raise concerns that 20ExM induces significant distortion in axial structures. The new mitochondria and brain images, while demonstrating micron/submicron scale preservation, do not convincingly prove the preservation of nanoscale architecture, which is a primary focus for most 20ExM users. Artificial distortion due to physical expansion and/or protein digestion remains a significant concern in ExM, especially with expansions larger than the original 4-fold. The authors must provide validated structures that ensure nanoscale structural preservation, which is the principal concept of 20ExM for a broad audience. This might be achievable using centrosome 9-cartwheel structures, among others. If antibody size is a concern, labeling can be performed by expressing tag-conjugated proteins such as GFP (enhanced by commercial nanobodies for GFP to minimize extra size), SNAP, etc.

Minor concern

The authors addressed the expected signal reductions due to 20ExM in a step-by-step response. However, this significant limitation could severely restrict its applications, as it will not work with typical primary-secondary antibody staining. While this limitation does not diminish the impact of the technology, the authors must clearly and extensively discuss these points in the Main text. Specifically, they should address the expected signal reductions, the necessity for tertiary antibody staining, the potential limitations, and possible ways to overcome these technical barriers. These aspects are crucial for readers and future users.

Authors' Response

Regarding the Reviewer's two major concerns, regarding preservation of nanoscale biological architecture, and distortion in axial structures, it is important to note that certain characterizations by the Reviewer stand in contrast to the published literature, and other concerns are easily addressed by thinking about the nature of the experiment at hand.

Regarding the images of the nuclear pore complex, the Reviewer argues that our images are unconvincing, due to the following two reasons:

1. "Many strong Nup96 puncta outside the circle marks are randomly distributed, and"
2. "the nuclear pores marked as circles by authors do not resemble known nuclear pores."

Response to Reason 1: As indicated in the figure legend, we circled some nuclear pore complexes, but not all nuclear pores in the figure. Many puncta outside of the circles are likely part of other nuclear pore complexes. Importantly, in earlier best-practices ExM studies visualizing nuclear pore complexes, even with specialized fixation, purification, and staining methods designed to optimize nuclear pore appearance, the investigators often observe single puncta that do not form a ring, such as in the figure on the right of panel **A**, below, taken from the highest-performing previous ExM paper that looks at nuclear pores. These puncta could

arise from non-specific staining, or from a lack of staining against a fraction of the Nup96 proteins within a given pore.

Response to Reason 2: We disagree with the Reviewer's characterization that nuclear pores marked with circles do not resemble known nuclear pores. As shown in the figure below comparing our NPC images (**A**, left) with recently published state-of-the-art iU-ExM images (**A**, right), achieved with a similar staining strategy (using one antibody, against a fluorescent protein attached to Nup96), but with additional methods designed to optimize nuclear pore appearance (e.g., nuclear extraction, PFA fixation at a different concentration), we observe highly similar appearance of NPCs.

A This work
4% PFA Fixation

iU-ExM (*Nat. Comm.* 2023) Fig. S2f
Nucleus extraction + 2.4% PFA
Fixation

B < 4 corners

≥ 4 corners

(A): **left**, our work (raw image without Gaussian filtering. brightness and contrast adjusted to highlight NPC corners); **right**, state-of-the-art previously published images of nuclear pores, recently published (iU-ExM, *Nature Communications* (2023) 14:7893, Fig. S2f)

(B): **top row**, our work, for nuclear pores with less than 4 corners visible (left) and with more than 4 corners visible (right).

Bottom row, the best previous nuclear pores seen with ExM, for nuclear pores with less than 4 corners visible (left) and more than 4 corners visible (right).

(C): our work: a box plot of NPC radius. (Extended Fig. 5b)

(D): “best previous” nuclear pore ExM paper: a scatter plot of NPC diameter (iU-ExM, Fig. 2i). Note the close match between our numbers and their numbers.

(E): our work: a histogram of the number of corners visible per NPC. (Extended Fig. 5c)

(F): “best previous” ExM paper: a histogram of number of corners visible per NPC (iU-ExM, Fig. S2i). Note similar fractional staining of pore proteins.

Scale bars, (A): Left, 250 nm; inset, 50 nm; Right, 240 nm. (B), 50 nm. All in biological units.

In short, our data match the best outcomes of previous ExM studies of the nuclear pore, that involve more elaborate expansion protocols. We note that our paper is not trying to exceed the performance of previous 20x expansion protocols, but only to create the most efficient way of achieving that performance - replacing laborious iterative expansion procedures, with a single, simple, step. The close match of our work, to the best-practices precedent, achieves that goal.

We tried to highlight these facts for the reviewer: as indicated in our previous **Response to Reviewers** (and associated Figure Legend), “We noticed some single-puncta signals that did not participate in a ring. We did not use a special NPC preparation strategy, as is common for microtubules. More specialized fixation methods, such as permeabilization with detergent prior to fixation, cryofixation with methanol, or extracting nuclei from intact cells prior to staining and expansion, might in principle further improve staining quality. Our goal in the current study was not to study NPCs, but rather to validate the resolution and gel-contributed error of 20ExM with NPCs. Furthermore, our goal was not to better earlier iterative methods like iExM or ExR, but rather to show that such performance could be achieved in a single step. Since our current protocol was sufficient for these purposes, we did not pursue further optimization.” These specialized methods were developed to selectively improve the appearance of NPCs, whereas we used a standard PFA fixation method, as used most commonly by users of ExM. Indeed, the method of extracting nuclei out of cells through centrifugation prior to fixation, staining, and expansion, as used in the iU-ExM study, does not resemble a typical ExM user’s application, nor is it representative of methods that optimally preserve biological ultrastructure. The method described in *Nat. Methods* 16, 1045–1053 (2019), could in principle improve the appearance of nuclear pores, by reducing single-puncta signals that do not participate in rings, but is highly specialized: the permeabilization protocol involves pre-fixation with 2.4% PFA, plasma membrane permeabilization with digitonin, a first round of staining, fixation with 3% PFA, nuclear envelope permeabilization with Triton X-100, a second round of staining, and final washes.

The Reviewer also mentioned that some microtubule images raised concerns, but did not elaborate. Our microtubule images are consistent with previous studies in both appearance, and measured thickness. Unlike NPCs, we have previous experience in preserving ultrastructure (with specialized fixation) of microtubules, since we collaborated with the lab of Xiaowei Zhuang on this in our 2017 *Nature Methods* paper on iExM (Hazen Babcock, of that lab, was an author). Our microtubule images indicate that with such ultrastructure preparation, 20ExM produces highly similar images as previous iterative expansion methods and STORM, as shown in this figure:

Microtubules: **Left**, our work (Fig. 2a, imaged in the middle of a cell, with microtubules entering and exiting the imaging plane); **right**, “best previous” microtubule ExM paper (iExM, Fig. 2d, imaged at the bottom of a cell; this leads to the appearance of longer microtubule segments, since they are flat and parallel to the bottom of the cell, and thus run for longer distances in the imaging plane)

Regarding z-axis distortion, the Reviewer pointed out that there are differences between pre- and post-expansion images, due to the following two reasons:

1. “There are many new signals in the post-20ExM images, and”
2. “shapes differ significantly, even in regions where both images show signals.”

However, the Reviewer’s comment that these differences indicate axial distortion introduced by 20ExM does not consider our experimental paradigm, and therefore is inaccurate. We encountered two challenges when performing z-axis distortion analysis, including:

1. While we made our best attempts to ensure samples are in the same orientation during confocal imaging before and after expansion, slight differences (less than 10° rotational difference, based on our visual examination) occurred. While miniscule, the slight rotation of the sample during confocal imaging before and after expansion resulted in a slight angle difference in the xz and yz plane that we select in the same sample’s pre- and post-expansion images. We found this challenging to correct computationally. Indeed, in almost all ExM papers that compare pre- and post-expansion images, there is

some different between the appearance of the two images, due to this aspect of the imaging process. It's simply a reality of imaging the same sample twice.

2. Because we use relatively thick brain tissue slices (50 μm thick), we observed a decrease in signal intensity at high-z positions (i.e., further away from the imaging lens, or the right side of the "YZ plane" images), due to scattering of light by lipids, whereas expanded (and therefore cleared) samples do not suffer from the same signal decrease. This resulted in apparent differences at high-z positions between pre- and post-expansion images.

Regarding Reason 1, the new signals in post-20ExM images are due to Challenge 2, that we described above. For example, the red-boxed area in the post-20ExM image is highly visible, whereas the boxed area in the pre-20ExM image only contains a faint signal. That is because this boxed area is at a deep z-depth, as far as confocal imaging is concerned. When we examine the same area in the xy-plane image at this z position, we observe the same neuron in pre and post-20ExM images. But, consistent with Challenge 2, in the "Pre XY plane" image, there is less signal than in the "Post XY plane" image. Consistent with Challenge 2, all of the reviewer's claimed "new signals in the post-20ExM images" are at high-z positions, where pre-20ExM images have low signal intensity due to confocal limitations in scattering tissue.

Pre YZ plane Post YZ plane Pre XY plane Post XY plane

Left two panels: pre- and post-expansion YZ plane images of the same sample at corresponding X positions. Towards the right, is deeper in the slice (farther from the objective lens of the confocal).

Right two panels: pre- and post-expansion XY plane images of the same sample at corresponding Z positions.

Red dotted boxes mark approximately the same region in all images. The "Pre YZ plane" and "Pre XY plane" images have the same brightness and contrast settings. The "Post YZ plane" and "Post XY plane" have the same brightness and contrast settings.

Regarding Reason 2, the overall shape of soma and dendrites are similar between pre- and post-20ExM images. The slight difference in shape is likely due to Challenge 1.

Regarding the Reviewer's suggestion to use centrosome 9-cartwheel structures, among others, as "validated structures that ensure nanoscale structural preservation," it is not clear to us how a structure like the centrosome can offer additional information beyond what we've validated with microtubule and NPCs. Centrosomes consist of nine microtubule structures, which are 1-2 μm away from each other (iU-ExM, Fig. 5d, *Nat. Commun.* 14, 7893 (2023); Magnify, Fig. 6, *Nat. Biotechnol.* 41, 858–869 (2023)), significantly above the 100 nm threshold for evaluating nanoscale biological structures. Yet, both microtubules (~60 nm peak-to-peak, as here stained) and NPCs (~40 nm corner-to-corner) structures are smaller than 100 nm and already count as nanoscale biological structures.

In summary, we think that we have shown our protocol to be the equal in performance to previous 20x protocols. We did not intend to exceed their performance, only to make the protocol practical for everyday biology. Thus, we do not think that any additional experiments are needed. That said, if the editors deem it necessary, we would be willing to consider doing additional experiments, as long as they are feasible.

Reviewer #1:

Remarks to the Author:

Super-resolution microscopy stands as a pivotal tool in cellular biology, with a persistent need for advancement to visualize previously unseen structures. The authors have introduced a new, optimized ExM method capable of a one-step 20-fold expansion. While ExM is already recognized as a cost-effective and robust super-resolution technique, the prevalent method offers only a 4-fold expansion. Achieving expansions greater than 5-fold typically requires multiple expansion steps or specialized equipment. This study introduces a one-step method that allows for a 20-fold expansion without any specialized equipment. In theory, the 20-fold expansion can attain an approximate lateral resolution of 12 nm when used with a standard confocal microscope, given the typical 240 nm lateral resolution of confocal microscopy. Supporting this, the authors noted an accuracy of roughly 17 nm in lateral measurements. I believe this 20ExM method can serve as a valuable tool, offering enhanced "native" resolution without the need for post-imaging processing across various research domains. I recommended this work for publication in the leading journal, Nature Methods, after addressing following concerns.

Major concerns

1. The authors demonstrated that gels with varying gelation times displayed different expansion rates (see Extended Figure 4). It's puzzling to note that none of the time points reached a 20x expansion, while gels containing biological specimens (tissue cells) achieved a 21-fold expansion. It would be beneficial if the authors could provide gel size measurements for both cells and tissues that align with the expansion rates observed in biological samples. Additionally, an insight into how gelation time influences these measurements with biological specimens and their reproducibility would be valuable.

1. Regarding "It's puzzling to note that none of the time points reached a 20x expansion, while gels containing biological specimens (tissue cells) achieved a 21-fold expansion": first, we observed that with identical initiator concentration, shorter gelation time leads to higher expansion factor in pure gels (Extended Fig. 4b). Second, we observed that the presence of a biological specimen, such as a cell culture, slowed the polymerization kinetics: "...the same gel formula, with a cell or tissue embedded, required a longer time to gelate, for a given targeted expansion factor, than a pure

gel.” Thus, although both pure gels and cell-culture containing gels use the same initiator concentration, because the presence of cell culture slows down polymerization, a 2-hour-polymerized cell culture achieved higher expansion factor (22x) than a 1-hour-polymerized pure gel (16x). For the tissue protocol, since we reduced the initiator concentration to ensure sufficient time for the monomer solution to diffuse into the tissue, the polymerization is further slowed due to lower initiator concentration. 16–20-hour-polymerized brain tissue achieved higher expansion factor (18x) than 1-hour-polymerized pure gels (16x). It is possible that shortening the gelation time for a pure gel to less than 1 hour could lead to larger expansion factors, such as 20x expansion. However, since our objective was to expand biological specimens, we did not pursue further optimization of the expansion of pure gels.

These observations are consistent with our understanding of the polymerization mechanism. Due to the DMAA-SA gel’s radical-dependent self-crosslinking mechanism (*Macromolecules* 2014, 47 (13), 4445–4452), longer polymerization times could lead to accumulated radical activity, and thus crosslinking, and thus lower expansion factors. Some polymer research groups have anecdotally informed us that coarse surfaces slow down polymerization of gels that touch the surface. Biological specimens introduce some degree of coarseness compared to pure glass surfaces, and thus could in principle slow down polymerization. However, as our emphasis is bioengineering and the biological applications of such inventions, as opposed to pure polymer chemistry, we did not investigate this further.

2. Regarding “gel size measurements for both cells and tissues that align with the expansion rates observed in biological samples”, we performed 20ExM on cells and tissues, using our standard protocols for each, to quantitatively measure the expansion factor, both by assessing physical gel size as requested, as well as by examining biological landmarks in pre- vs. post-expansion samples, as we had previously done.
 - a. For cells, we now provide physical gel size measurements for two of the four HEK293 cell batches stained with anti-beta-tubulin antibodies and then expanded, that we reported in the initial

manuscript (we did not measure physical gel size for the other two cell batches). At the time, we had measured the size of each gel with a ruler immediately after gelation, and then again after full expansion. We observed 22.4 ± 1.0 (mean \pm standard deviation)-fold expansion when physical gel size was assessed, vs. 22.3 ± 0.8 -fold expansion when biological landmarks were utilized, for these two cell culture batches.

- b. For tissue, we performed 20ExM with Thy1-YFP transgenic mouse brain slices, using the tissue protocol. Pre-expansion, in order to use biological landmarks to calculate expansion factor, we imaged YFP signals with confocal microscopy before gelation; to measure physical gel size, we used a ruler, after gelation. Post-expansion (in more detail: we performed softening, anti-GFP staining, and expansion), we re-measured the physical size of the gel with a ruler, and re-imaged the same region in the brain slice with a confocal microscope to use biological landmarks to calculate expansion factor ($n = 2$ brain slices from 1 Thy1-YFP transgenic mouse). We observed 18.5 ± 1.1 (mean \pm standard deviation)-fold expansion when physical gel size was assessed, and 19.0 ± 0.7 -fold expansion when biological landmarks were utilized.

Thus, the physical gel size-assessed expansion factor for both cells and tissues matched the expansion factor assessed when biological landmarks were used. This has now been added to Supplementary Note 2.

3. Regarding “how gelation time influences these measurements with biological specimens and their reproducibility”, we used Thy1-YFP transgenic mouse brain slices, and performed 20ExM, in brain tissue protocol form, with varying gelation times (6, 16–20 (standard, the same samples used in Author’s Response 2b), and 72 hours; $n = 2$ brain slices from 1 mouse for each condition). We found that 6 hours was not sufficient to complete gelation (i.e., the gel didn’t fully polymerize), whereas 16–20-hour gelation samples expanded 18-fold (see statistics above, in Author’s Response 2b), consistent with the 2 previous brain slices we reported in the initial manuscript. The 72-hour gelation samples expanded 10.1 ± 0.3

(mean \pm standard deviation)-fold. This has now been added to Supplementary Note 3.

2. Related to 1, authors should provide expansion rates in z-axis of gels and biological specimen.

4. To “provide expansion rates in z-axis of gels and biological specimen,” we quantitatively measured the z-axis expansion factor of gel-embedded brain tissue (the same samples used in Author’s Response 2b) by measuring physical gel size, as well as by utilizing biological landmarks, pre- vs post-expansion, with a focus on the z-axis:

- a. For physical gel size, we measured pre- and post-expansion gel thickness with a confocal microscope, and obtained the expansion factor. In particular, we focused on the gel-adjacent surfaces of parafilm spacers (which flank the gel closely and were more autofluorescent, and thus more visible by the confocal, than the gel itself), to determine pre-expansion thickness, and on the expanded gel top and bottom, to determine post-expansion thickness. We observed 18.0 ± 0.4 (mean \pm standard deviation; $n = 2$ brain slices from 1 Thy1-YFP transgenic mouse)-fold z-axis expansion, as assessed by physical gel size.
- b. We measured pre- and post-expansion distances between the highest and lowest (along the z-axis) visible YFP signals in the slice (we used 50- μm thick Thy1-YFP brain specimens), to serve as biological landmarks, to calculate expansion factor. We observed 18.2 ± 0.5 -fold z-axis expansion via analysis of these biological landmark signals (mean \pm standard deviation; $n = 2$ brain slices from 1 Thy1-YFP transgenic mouse).

These z-axis expansion factors are consistent across the two different methods of measurement, and are also consistent with the xy-plane expansion factors measured above for a gel specimen containing tissue (~18-fold; see Author’s Response 2). This has now been added to Supplementary Note 2.

3. An advantage of 20xExM is its rapid gel expansion, in contrast to the 10xExM gels which necessitate a lengthier expansion process involving several water exchanges. We observed that 10xExM gels begin to contract after reaching a 10x expansion, typically after a few hours. It would be valuable if the authors could provide information regarding the stability of the expanded gels and suggest an optimal timeline for imaging to ensure consistent expansion.

5. To “provide information regarding the stability of the expanded gels,” we examined the size of an expanded brain-tissue-embedded gel at 5 minutes, and at 2, 21, and 25 hours, after full expansion was achieved, within a capped imaging plate (n = 1 gel). We found that the gel did not visibly contract or exhibit other obvious changes over the course of 25 hours (Supp. Fig. 4). In addition, in all the aforementioned studies, consistent expansion factors, with small standard deviations, were observed, without particular attention to timing. Thus, especially in a humidity-controlled environment, gels may be stable over the course of a day or so. This has now been added to Supplementary Note 4.

4. The authors utilized the microtubule as a cellular ruler, a fitting choice given the well-studied nature of microtubule structures. However, the measured thickness of microtubules labeled with primary and secondary antibodies, as captured by super-resolution microscopy, ranged from 35-70 nm. This raises questions about the actual sizes of antibody-labeled microtubules. While it's not imperative for this study to validate the true thickness of the microtubules, the authors should offer both expansion factor measurements and error estimates with multiple rulers. To enhance reliability, they might consider using another well-characterized cellular ruler, such as the centrosome.

6. Regarding “the measured thickness of microtubules labeled with primary and secondary antibodies, as captured by super-resolution microscopy, ranged from 35–70 nm. This raises questions about the actual sizes of antibody-labeled microtubules”: antibody-labeled microtubules have been extensively imaged and commonly used as a standard by the super-resolution community, having been imaged with STORM, STED, ExM, and many other methods (e.g., *Science* **2007**, 317 (5845), 1749–1753; *Nanoscale* **2018**, 10 (37), 17552–17556). For example, in our previous study on iterative expansion with 20x magnification (iExM, *Nat. Methods*

2017, *14* (6), 593–599), STORM images of primary antibody-stained microtubules resulted in a width that ranged from 25 to 50 nm, with a mean and standard deviation of 37.3 nm and 4.7 nm respectively. The iExM images of primary and secondary antibody (bearing DNA oligos for amplification of brightness)-stained microtubules resulted in a range of 25 to 90 nm, with a mean and standard deviation of 58.7 and 10.3 nm respectively. The latter number, in particular, could be regarded as an upper bound (because it includes any real biological variability in microtubule thickness) on the nanoscale error introduced by iExM. It has been modeled and observed that primary antibody-labeled microtubules have an average diameter around 40 nm, and primary and secondary antibody-labeled microtubules have an average diameter around 60 nm (*Science* **2007**, *317* (5845), 1749–1753; iExM, *Nat. Methods* **2017**, *14* (6), 593–599; *Nanoscale* **2018**, *10* (37), 17552–17556; *EMBO Rep.* **2018**, *19* (9). <https://doi.org/10.15252/embr.201845836>), consistent with the iExM measure. Our measurement of 100 microtubule diameters resulted in an average of 62.7 nm and standard deviation of 8.8 nm, which compares well to those reported by the original iExM protocol, and suggesting a similar high resolution and low distortion of 20ExM. This discussion has now been added to Supplementary Note 5.

7. Following the Reviewer's suggestion of “using another well-characterized cellular ruler,” we explored the nuclear pore complex (NPC) as an alternative ruler, which has been imaged with ExM, ExM-SIM, ExM-SRRF, TREx, and iU-ExM to validate their technologies (*Nat. Methods* **2019**, *16* (10), 1045–1053; TREx, *Elife* **2022**, *11*. <https://doi.org/10.7554/eLife.73775>; iU-ExM, *Nat. Commun.* **2023**, *14* (1), 7893). Note that in contrast to the fixation and labeling protocol for microtubules in cells (in summary: applying extraction buffer, tubulin fixation buffer, reduction solution, and quenching buffer, followed by primary and secondary antibody staining), which has helped microtubules to be used as a benchmark for resolution in various studies, by supporting the obtaining of very clean signals and low background, we found that a variety of fixation and labeling methods were used for NPCs in different studies, with some methods extracting nuclei from intact cells prior to staining and expansion. We thus used standard 4% paraformaldehyde to fix intact *NUP96::Neon-AID* DLD-1 cells, where a nuclear pore protein

NUP96 was fused to a fluorescent protein, mNeonGreen (*bioRxiv*, **2020**, 2020.11.13.381947; *Nature* **2021**, 598 (7882), 667–671) and then stained with anti-mNeonGreen antibodies. We performed 20ExM and imaged NPCs on the top and bottom of the nuclei which are tangential to the imaging plane, critical for seeing the circular shape of the nuclear pore in the imaging plane. We then measured the NPC radius and number of corners as done in the aforementioned studies. This has now been added to the Main Text as well as Extended Fig. 5.

- a. We manually picked NPCs with at least 4 visible corners in top view after max intensity z projection, and measured the NPC radius, via radial intensity profiling, obtaining a radius of 55.4 ± 8.9 nm (mean \pm standard deviation; $n = 35$ NPCs from 2 culture batches; expansion factor = 22.8 ± 0.4 as measured by gel size), consistent with the expected radius of 53.5 nm based on previously reported cryo-EM structures (Extended Fig. 5a,b) and previous expansion microscopy studies (*Elife* **2022**, 11. <https://doi.org/10.7554/eLife.73775>; *Nat. Commun.* **2023**, 14 (1), 7893), and exhibiting an expansion error upper bound (the aforementioned standard deviation of 8.9 nm) comparable to that measured using microtubule diameter, above, for both 20ExM and iExM.
- b. A NPC ring consists of 8 individual corners, which are ~ 42 nm apart from each other, according to previous cryo-EM data. 20ExM clearly resolved individual corners, similar to what has been observed with dSTORM and iU-ExM (*Nat. Methods* **2019**, 16 (10), 1045–1053; *Nat. Commun.* **2023**, 14 (1), 7893). In addition, consistent with a previous study, we automatically counted the corners of manually selected NPCs using the “Counting Corners” algorithm reported in the iU-ExM paper, and observed a similar distribution of numbers of corners per NPC as in the previous study (Extended Fig. 5c; source data for comparison: Fig. S2i from iU-ExM, *Nat. Commun.* **2023**, 14 (1), 7893). We then measured the distance of adjacent corners as determined by the “Counting Corners” algorithm, and determined the distance to be 48.6 ± 12.8 nm (Extended Fig. 5d; mean \pm standard deviation; $n = 108$ measurements of 35 NPCs from 2 culture batches), consistent with the expected distance, and exhibiting an

expansion error upper bound (the aforementioned standard deviation of 12.8 nm) comparable to that measured using microtubule diameter and NPC radius.

- c. We noticed some single-puncta signals that did not participate in a ring. As noted above, we did not use a special NPC preparation strategy, as is common for microtubules. More specialized fixation methods, such as permeabilization with detergent prior to fixation, cryofixation with methanol, or extracting nuclei from intact cells prior to staining and expansion, might in principle further improve staining quality (*Nat. Methods* **2019**, *16* (10), 1045–1053; *Nat. Commun.* **2023**, *14* (1), 7893). Our goal in the current study was not to study NPCs, but rather to validate the resolution and gel-contributed error of 20ExM with NPCs. Furthermore, our goal was not to beat earlier technologies like iExM or ExR, but rather to show that such performance could be achieved in a single step. Since our current protocol was sufficient for these purposes, we did not pursue further optimization.

In conclusion, 20ExM can reliably achieve low errors and good resolutions, comparable to previous state of the art protocols, supported by characterization of multiple independent rulers.

5. Figures I and J. FRC measurements are influenced by the signal-to-noise (S/N) ratio. It would be beneficial for the authors to include intensity data alongside these results. While pinpointing the precise maximum resolution isn't crucial, it would be enlightening to see structures that approach this estimated maximum resolution. If 20ExM yields a 17 nm lateral resolution, actin might serve as a more appropriate ruler compared to microtubules for 20ExM.

8. Regarding “FRC measurements are influenced by the signal-to-noise (S/N) ratio. It would be beneficial for the authors to include intensity data alongside these results”:

- a. We conducted line intensity profile analyses on microtubule images generated using 20ExM or iExM protocols, using previously published iExM data (since the original iExM protocol is not in much use anymore, with the ExR protocol having largely replaced it). Our

findings revealed that the 20ExM protocol yielded similar-appearing images, and line profiles, between 20ExM and iExM (Supp. Fig. 2b,c).

- b. In principle, noise in confocal images could originate from multiple sources, including the immunohistochemistry protocol itself (e.g., non-specific binding), and imaging shot noise. We attempted to keep staining protocol noise as small as possible, by strictly following a microtubule staining protocol (i.e., extraction, fixation, etc.) used in previous studies for measuring resolution (*Science* **2007**, 317 (5845), 1749–1753; *Nat. Methods* **2017**, 14 (6), 593–599). To assess the impact of shot noise, we performed FRC analysis on the same image pair both with and without Gaussian filtering, which reduces shot noise (*IEEE Trans. Biomed. Eng.* **2000**, 47 (12), 1600–1609; *Int. J. Biochem. Cell Biol.* **2021**, 140, 106077). We used a sigma value of 0.5 for the Gaussian filter to avoid blurring the signal too much, which may affect the resolution. Such Gaussian filtering reduces noise visibly, but the Global FRC barely changed – from 21.2 nm for non-Gaussian-filtered images, to 20.1 nm for Gaussian-filtered images (Supp. Fig. 1). Thus our FRC measurements for 20ExM remained consistent, regardless of shot noise.
- c. Additionally, we performed SNR analysis (calculated by dividing the signal intensity by the standard deviation of the background) on synaptic puncta as we did previously for expansion revealing (ExR). In particular, we analyzed SNR of synapses that were identified based on joint RIM1/2 and PSD95 presence, in post-expansion antibody stained 20ExM brain tissue (same images as in Fig. 3b) and found that the SNR was ~35 (Supp. Fig. 2d). Although we were not able to use the same primary and secondary antibodies that were used for the post-expansion antibody staining ExR paper (the RIM1/2 primary antibody used in the ExR paper was discontinued), the SNR of synapses that were identified through staining of Bassoon, Cav2.1, Homer1, PSD95, RIM1/2, Shank3, and SynGAP with the ExR protocol was on average ~15 (just to get a ballpark estimate, we averaged the SNR across all antibodies, using previously published data; source data: Extended Fig. 2d from the

ExR paper, *Nat. Biomed. Eng.* **2022**, 6 (9), 1057–1073). Thus we estimate that our SNR is comparable to those of earlier technologies, with our current goal simply being to make the process easier, by enabling it to occur in a single step, rather than requiring repeated steps.

This has now been added to Supplementary Note 6.

9. Regarding “While pinpointing the precise maximum resolution isn't crucial, it would be enlightening to see structures that approach this estimated maximum resolution”: as we mentioned in the initial manuscript, 20ExM revealed the hollow structure of microtubules. According to previous work, visualizing the hollow structure of microtubules requires at least 15x expansion, equivalent to 16–26 nm effective resolution (Fig. S2 from X10, *EMBO Rep.* **2018**, 19 (9). <https://doi.org/10.15252/embr.201845836>). We have also added nuclear pore complex (NPC) images (see Author's Response 7) where 20ExM resolved individual corners of NPCs which are around 42 nm apart from each other based on previous cryo-EM data. Furthermore, 20ExM images of RIM1/2 and PSD95 also revealed that synaptic nanocolumns align with each other with 20–25-nm precision (Fig. 3e,f), also approaching the measured resolution, and matching the precision characterized by expansion revealing (see Fig. 4i and 4k of that earlier paper). Finally, the estimated nanoscale error introduced by expansion was about 10 nm, consistent with that of iExM (see Author's Response 6), suggesting that for things larger than this size, the expansion factor - in this case, ~20x - should determine the resolution (300 nm / 20 ~ 15 nm). This has now been added to Supplementary Note 7.
10. Regarding “actin might serve as a more appropriate ruler compared to microtubules for 20ExM”: the common actin stain, phalloidin, does not carry primary amines with appropriate pKa to allow for direct anchoring to the expandable hydrogel, and subsequent post-expansion amplification. ExM with pre-expansion antibody staining introduces linkage error and increases effective diameter (*ACS Nano* **2020**, 14 (11), 14999–15010; *J. Vis. Exp.* **2021**, No. 170. <https://doi.org/10.3791/62079>). While actin is thinner than a microtubule (7 nm vs 25 nm), the expected peak-to-peak distance of actin after pre-expansion staining would be around 40 nm

(since pre-expansion primary and secondary antibody staining would introduce ~15 nm linkage error in each direction), larger than the expected error of 20ExM, and not too much smaller than the expected size of a labeled microtubule. Indeed, both situations are dominated by antibody size. One method used functionalized phalloidin that could be anchored to the gel to visualize actin with ExM (*ACS Nano* **2020**, *14* (7), 7860–7867). However, post-expansion imaging seemed to us to be quite challenging, since the 8000-fold volume expansion of 20ExM would require significant signal amplification to visualize something as narrow as an actin filament, and the functionalized phalloidin presented limited epitopes and functional groups for selective amplification. Thus, good visualization of actin with 20ExM may require better methods of labeling actin. Perhaps it is not surprising that entire papers have been devoted to the topic of ExM visualization of actin, with titles like “Trifunctional Linkers Enable Improved Visualization of Actin by Expansion Microscopy,” “Super-Resolution Three-Dimensional Imaging of Actin Filaments in Cultured Cells and the Brain via Expansion Microscopy,” and “Simple methods for quantifying super-resolved cortical actin.” This would make for a great downstream effort. That said, we are confident that our estimates of resolution are solid, as noted in Authors’ Response notes 8 and 9, above, and related text.

6. The microtubule images appear faint, likely due to the 20-fold enlargement of the stained samples. It would be valuable for the authors to describe a method for achieving high signal-to-noise (S/N) staining and also to provide quantitative measurements of the signal reduction.

11. Regarding “The microtubule images appear faint, likely due to the 20-fold enlargement of the stained samples” and “also to provide quantitative measurements of the signal reduction”: we agree that 20-fold enlargement of a stained sample will dilute signal intensity greatly. Indeed, after 20-fold expansion, we expect an 8000-fold increase in volume, and corresponding 8000x decrease in fluorophore concentration. As the pre-expansion raw signal intensity of cell culture microtubule staining ranged from 15000 to 30000 (Supp. Fig. 2a), we would expect the intensity to drop to ~2–4 after 8000-fold volumetric dilution. Indeed, after expansion, we could not observe any remaining fluorescence without amplification. This

has now been added to Supplementary Note 6.

12. Regarding “It would be valuable for the authors to describe a method for achieving high signal-to-noise (S/N) staining”, with amplification, after tertiary antibody staining, we observed post-expansion signal intensity to be above 200, a ~100-fold increase in signal intensity compared to the expected diluted intensity (Supp. Fig. 2b; note that Supp. Fig. 2a and 2b were acquired under identical microscope settings and processed identically). The signal intensity was sufficient to reveal clear hollow microtubule structures, and support distortion and resolution analyses (see Authors’ Response 6 and 8). We used post-expansion antibody staining to achieve sufficient SNR for our purposes (see Authors’ Response 8 and 11). We note that SNR could, in principle, be further improved with any one of a number of previously published signal amplification methods, such as hybridization chain reaction (HCR) and rolling circle amplification (RCA), which, as modular DNA-based methods, have easily been incorporated into ExM protocols by multiple groups. This has now been added to Supplementary Note 6.

7. 8. Similarly to point 7, the RIM1/2 and PSD95 staining images in 20ExM appear faint due to noticeable noise, even though the regions selected by the authors seem clear. This raises concerns about potential bias in measurements based on the regions chosen for quantification. The authors should offer quantitative data on signal intensity and discuss any challenges therein. Moreover, the images in Fig. 3b appear to have excessive contrast, making them look oversaturated.

13. Regarding “the RIM1/2 and PSD95 staining images in 20ExM appear faint due to noticeable noise, even though the regions selected by the authors seem clear”, the apparent difference in noise between the low-magnification and zoomed-in images was due to the z-projections of these images being conducted over different depths. Specifically, the low-magnification image was z-projected across the entire z range of the slice, whereas the zoomed-in images only contained ranges that contained a particular synapse. We have demonstrated this difference by showing the same synapses z-projected over the whole imaging range vs. the synapse-limited range, in Supp. Fig. 2f. This has now been noted in Supplementary

Note 6.

14. Regarding “This raises concerns about potential bias in measurements based on the regions chosen for quantification. The authors should offer quantitative data on signal intensity and discuss any challenges therein. Moreover, the images in Fig. 3b appear to have excessive contrast, making them look oversaturated.”: in the nanocolumn analysis, synapses were chosen based on the juxtaposition of RIM1/2 and PSD95 signals, as previously utilized in the ExR study. Signal-to-noise ratio (SNR) measurements, conducted as in the ExR study, revealed an SNR for 20ExM comparable to that of ExR (see Author’s Response 8), with both studies focused on synaptic proteins that are known to participate in nanocolumns. For Fig. 3b, we increased contrast to highlight the boundary of the synapses, not uncommon for studies emphasizing synaptic protein density shape, so we could easily identify synapses for subsequent data analysis. This does, notably, lead to many pixels within the synapses appearing saturated. We now also include the same images but with contrast adjusted to only have 1 pixel saturated per channel per image, in Supp. Fig. 2e, highlighting the internal heterogeneity within the signal distribution of RIM1/2 and PSD95. While the synapse protein gap, and the synaptic protein density shapes, were qualitatively similar, different contrast adjustments will of course emphasize different aspects of the data. We have updated the legend of Fig. 3b to include the following description: “Brightness and contrast settings: first set by the Fiji’s auto-scaling function and then manually adjusted to improve contrast and highlight the boundary of synapses; quantitative analysis in c–f was conducted on raw image data.” This has now been noted in Supplementary Note 6.

8. A concern in 20-fold expansion is isotropy in Z-axis. If authors can validate the isotropic expansion in Z-axis using biological specimens, it would be valuable.

15. Regarding “validate the isotropic expansion in Z-axis using biological specimens,” we imaged pre-expansion tissue with a 40x lens and post-expansion tissue with a 4x lens to aim for similar fields of view for downstream registration. We then performed distortion analysis across z-

depths, comparing pre- and post-expansion tissue. We observed ~5% distortion over a distance of 15 μm , consistent with the distortion observed in the xy-plane (Supp. Fig. 3). This has now been added to Supplementary Note 8.

Minor concerns

9. DMAA gels are notably softer than the MBA-based 4x ExM gels, and likely requires training in handling. It would be beneficial if the authors could share recommendations and best practices for handling and imaging these gels.

16. We did not observe expanded DMAA gels, when made according to our protocol, to be softer than expanded classical (called by the reviewer MBA-based) 4x ExM gels. Upon completion of gelation, DMAA gels were in our hands robust and elastic, as had been shown by their original discoverers (*Macromolecules* **2014**, 47 (13), 4445–4452), who remark upon the extraordinary mechanical properties of this gel. But, it is always useful to share best practices. As with expanded 4x ExM gels, we suggest gentle gel handling (e.g., transferring the gel from one plate to another, flipping the gel, etc., using specific strategies, as described at length in *Curr. Protoc. Cell Biol.* **2018**, 80 (1), e56). If desired, 20ExM gels can be transferred, or flipped, after shrinking the gel with 1 \times PBS (which results in a shrinkage, empirically speaking, to ~4x). Typically, after fully expanding in double deionized water, in an imaging plate, we remove excess water with transfer pipets and kimwipes, and then transport the imaging plate to the microscope. We then image the gel, while still in that imaging plate. We avoid transferring the expanded gel from one plate to another, or flipping the gel, while the gel is fully expanded (due to its large size). To transfer or flip the gel, we advise to shrink the gel by incubating in 1 \times PBS for 20 minutes, as described above, then transferring or flipping the gel as described in our 2018 protocol paper, and then re-expanding in water (which takes another hour).

We have updated Supplementary Note 1 to include these step-by-step instructions on handling and expanding 20ExM gels.

Reviewer #2:

Remarks to the Author:

In this manuscript, Wang et. al. have developed a single-shot expansion microscopy approach with a 20X expansion factor, and a working resolution of 18 nm. This end was achieved through optimizing the ratio of hydrogel components used for 10X expansion, N,N-dimethylacrylamide (DMAA) and sodium acrylate. The authors further manipulated gelation conditions by deoxygenating the gelation solution with nitrogen gas and working in an oxygen free environment. By tuning the potassium persulfate initiator concentration, as well as the gelation time, the authors were able to show convincing microtubule expansion in cell culture, as well neuron synapse architecture of mouse brain tissue.

A one-shot 20X ExM protocol represents a significant advance in expansion microscopy, completely avoiding the cumbersome protocols of iterative ExM, while reaching new ceilings in expansion factor. The additional ability to perform post-expansion labeling is an asset. This is especially highlighted by the visualization of the microtubule lumen, a remarkable feat in super resolution imaging. 20ExM seems relatively straightforward to perform and has the potential to become among the most utilized protocols for large-expansion factor imaging. If the following concerns are addressed, this work merits publication in Nature Methods.

Major Concerns

Validation of 20ExM on more than one organelle and tissue type. In the studies often cited, and previously published in some cases, by the authors in this manuscript (TrEX, 10X, Magnify, Expansion Revealing) multiple cell compartments, and with diverse antibody stains, were described to show proof of concept. The microtubule staining is remarkable, but it is important to see more examples of the protocol in action, on more diverse organelles, especially an example of a membrane-bound organelle.

17. Regarding “The microtubule staining is remarkable, but it is important to see more examples of the protocol in action, on more diverse organelles, especially an example of a membrane-bound organelle”, we now have included 20ExM images of mitochondria and the nuclear pore complex (Extended Fig. 5a-e). We observed the structure of mitochondria, upon labeling a protein, TOM20, found on its outer membrane, as well as

the geometry of the nuclear pore complex, upon labeling fluorescent proteins attached to the protein Nup96 (see Author's Response 7). This has now been added to Main Text.

Extended Figures 2,3, and 4 are not very intuitive. The methods and the text clearly describe how to perform the protocol. However, the extended figures made to visualize the experimental setup are lacking clarity. In Extended Figure 2, a better example of the deoxygenated glove-bag environment would be useful. In Extended Figure 3, it was unclear what divided panels a,b and c from panels d,e, and f. The illustrated schemes make the addition of the gelation solution more understandable, but the figure could benefit from clearer images. It is difficult to gauge the expansion factor in Extended Figure 4 without scale bars and being unable to read the ruler. The scaling of the individual gels also seems different.

18. We have updated the Extended Figures to address these concerns. Amongst other edits: we have replaced the photo of the glovebag, updated Supplementary Note 1 to provide more detailed instructions, updated Extended Fig. 3 with clearer images, and added scale bars to Extended Fig. 4.

Minor Concerns / Questions

An important aspect of this protocol, as well as all expansion protocols, are the ease and feasibility of setting it up in the laboratory. There is a non-trivial difference between the 4 uL of initiator being added to the brain slice, and 20 uL being added to cultured cell. This is true as well for the 16-20 hours of gelation time for brain slices, and 2 hours for cultured cells. Have the authors further tested sections of different tissues to see if the initiator concentration or gelation time will need further adjustments, or do they expect that the 4 uL /20-hour 20 uL / 4-hour benchmarks will serve a diverse spectrum of tissues, or do you expect modifications of these ratios to ensure maximum expansion?

19. Regarding "Have the authors further tested sections of different tissues to see if the initiator concentration or gelation time will need further adjustments, or do they expect that the 4 uL /20-hour 20 uL / 4-hour benchmarks will serve a diverse spectrum of tissues, or do you expect

modifications of these ratios to ensure maximum expansion?": we have tested kidney and spleen sections with the protocol for brain slices (4 μ L/16–20 hour). We found that with SDS softening at 95 °C for 1 hour, gels containing kidney and spleen tissue became distorted and folded. This is consistent with our observation that tissues that are more fibrous than brain may require stronger softening than with heat and detergent alone (e.g., see our most recent paper on this topic, *Sci. Transl. Med.* **2024**, *16* (732), eabo0049). That said, alternative and stronger softening methods, using enzymes, work just fine. With LysC/Trypsin digestion in 1 mM EDTA, 50 mM Tris-HCl pH 8 and 0.1 M NaCl buffer at 37 °C overnight, as reported in the uniExM paper (*PLoS One* **2023**, *18* (9), e0291506), for example, both kidney and spleen reached 16.5 ± 0.4 fold expansion (mean \pm standard deviation; Extended Fig. 5f–h; n = 2 spleen, 2 kidney sections from 1 mouse measured by gel size), slightly less than brain tissue expanded under its corresponding 20ExM protocol. Thus, whereas 4 μ L/16–20-hour gelation may generally work across tissue types, tissues with more challenging mechanical properties may require harsher softening methods. This has now been added to the Main Text and Extended Fig. 5.

Some of the references to the Extended Figures are incorrect in the Supplementary Note. For example, Line 49 refers to sealing the connection between the tube and the glove bag with tape, which is Extended Figure 2b, and lines 90-91 incorrectly attributes adding nitrogen to the gelation solution to Extended Figure 4a.

20. We have addressed these inconsistencies.

Reviewer #3:

Remarks to the Author:

The manuscript by Wang et al describes a novel polymer formulation for performing expansion microscopy. The authors provide a hypothesis that explains why atmospheric oxygen may prevent uniform polymerisation and, hence, uniform expansion of the sample using this polymer. They show that by using an inert atmosphere (in a disposable glove bag) the polymerisation reaction can be carried out robustly and that this combination of methodological

development, along with the novel formulation enable unprecedented expansion factors from a single polymerisation reaction.

The manuscript is technically sound and sufficient information is provided to reproduce the work. Indeed, the Supplementary photos and descriptions are extremely useful in guiding the practical implementation of the method, a feature which is often overlooked.

In its current form, the manuscript does not show meaningful application to a biological question and doesn't, therefore, demonstrate any significant advantage over existing approaches for super-resolution imaging. For example, are the images significantly improved (by some metric) compared with iExM, what is the need for 20x over 10x expansion factor?

21. Regarding significant advantage, over existing 10x ExM approaches: all 10x expansion protocols to our knowledge (i.e., MAGNIFY, *Nat. Biotechnol.* **2023**, 41 (6), 858–869; TReX, *Elife* **2022**, 11. <https://doi.org/10.7554/eLife.73775>; X10, *EMBO Rep.* **2018**, 19 (9). <https://doi.org/10.15252/embr.201845836>) struggle to resolve the microtubule lumen on conventional microscopes, with the expected peak-to-peak distance, when pre-expansion primary and secondary antibody staining are used (Supplementary Table 1). One previous study conducted modeling which suggested that at least 15x expansion was needed to clearly visualize the microtubule lumen (Fig. S2 from X10, *EMBO Rep.* **2018**, 19 (9). <https://doi.org/10.15252/embr.201845836>). Only 20x methods, previously only achievable through iterative expansion methods such as iExM, could clearly reveal the microtubule lumen.

Please note - our goal, for 20ExM, was not to offer improved resolution over previous 20x iterated ExM protocols (e.g., iExM, ExR), but instead to enable 20x expansion to occur in a single step, rather than requiring multiple steps, so that 20x expansion would be faster and easier to perform.

Indeed, despite iterated ExM protocols having been published in multiple papers by our group and other groups (*Nat. Methods* **2017**, 14 (6), 593–599; *Nat. Commun.* **2020**, 11 (1), 3850.; *Nat Biomed Eng* **2022**, 6 (9), 1057–1073.; *Nat. Commun.* **2023**, 14 (1), 7893.), and having been cited ~500 times, none of these protocols are in widespread use in biology, the

way that, say, the single-step 4x proExM or MAP protocols are popular. Clearly, ease of use is important, and for 20x expansion microscopy to enter routine use, it would be useful for it to become easier to perform. The rapid spread of the single-shot 10x protocols mentioned in the previous paragraph suggests that if a single-shot 20x protocol were published, it might make 20x expansion commonplace. Indeed, in our own lab, there is great interest in using 20ExM, and prospective beta testers have also expressed enthusiasm. In summary, our goal is not to exceed the performance of the previous best-resolution expansion chemistries, but simply to make such expansion easier to achieve.

22. We used 20ExM to visualize synaptic nanocolumns in mouse brain tissue, in which pre- and post-synaptic proteins (e.g., RIM1/2 and PSD95, respectively) align with each other with ~20–25-nm precision (Fig. 3e,f). To our knowledge, no 10x ExM protocol has visualized synaptic nanocolumns, on conventional microscopes. Only 20x methods, previously only achievable through iterative expansion, such as ExR, were used to visualize such nanocolumnar alignment.
23. We have imaged nuclear pore complexes and observed the geometry of individual corners within a ring, which are ~42 nm apart from each other. Previous 10x methods, such as TREx, struggled to visualize individual corners (Fig. 3D from TREx, *Elife* **2022**, *11*. <https://doi.org/10.7554/eLife.73775>). Only the iterative 16x method iU-ExM has demonstrated the visualization of these corners, amongst all expansion protocols published to date (iU-ExM, *Nat. Commun.* **2023**, *14* (1), 7893). As noted above, our goal is not to exceed the performance of iterative expansion methods, but simply to enable that same high performance, in a single step.

A significant challenge with expansion microscopy is final image quality and particularly as seen for continuous features such as microtubules, where labelling can be punctate and patchy. It would be useful to see more on this in the manuscript- naively, one might expect a single polymerisation reaction to lead to some improvement in labelling/image quality (less fluorophore/antigen degradation).

24. Regarding “It would be useful to see more on this in the manuscript-naively, one might expect a single polymerisation reaction to lead to some improvement in labelling/image quality (less fluorophore/antigen degradation)”: we observed similar performance in labelling quality between 20ExM and ExR when we compared, side-by-side, the proteins PSD95 and RIM1/2 (see Author’s Response 8). We note that both of these datasets used post-expansion staining, where fluorescent antibodies are applied after polymerization, thus avoiding fluorophore degradation known to occur during free radical polymerization. It is possible that antigen degradation does not worsen that much, over 1 vs. 2 rounds of expansion. But, as noted, our goal in this paper was not to exceed the performance of iterative 20-fold expansion methods, but simply to show that such excellent performance could be achieved in a single step.

A general comment about the narrative around absolute distance measurements in the manuscript. The authors quote a standard deviation in the expansion factor of approximately 8% (21.5 +/- 1.7x). As I understand, in the manuscript, expansion factor is calculated for each gel, with pre- and post-expansion imaging. I think it’s unlikely that the majority of users will consider using pre-expansion imaging in this way, and indeed, may lack the expertise to perform the registration of pre-/post-expansion images and accurately derive the expansion factor for a given gel. It should be clear from the manuscript that such an approach is necessary to perform quantitative measurement in the samples.

25. Many groups perform a pre-ExM low-magnification check of overall sample size and/or the dimensions of key features, for comparison to post-expansion measurements of the same features, and thus expansion factor calculation. To calculate expansion factor, please note that registration is not needed. Instead, users can simply measure the distance from one boundary of the sample to another, or from one biological landmark to another, and then compare that measurement between pre- and post-expansion states, as previously reported for earlier ExM methods (*Science* **2015**, 347 (6221), 543–548, *Nat. Biotechnol.* **2016**, 34 (9), 987–992). We believe this kind of measurement is necessary for quantitative measurement, and we have provided instructions in our 2018 protocols paper (*Curr. Protoc. Cell Biol.* **2018**, 80 (1), e56). This has now been added to Supplementary Note 9.

Reviewer #1:

Remarks to the Author:

Expansion Microscopy (ExM) has become an essential toolkit for broad applications in cell and developmental biology, offering super-resolution capabilities to researchers without the need for expensive super-resolution microscopes. Current ExM techniques are limited to 4-10-fold expansion, with greater expansions typically requiring iterative processes that are time-consuming and complex. This new 20ExM method improves the chemistry to enable a one-step process, achieving a 20-fold expansion. Although the use of ExM is significantly increasing worldwide, a major concern and limitation of these methods is the preservation of biological architectures due to the physical expansion and protein digestion processes. Particularly, the 20-fold expansion raises serious concerns, and strong evidence must be provided to ensure the preservation of biological structures. The significant impact of the 20ExM method lies in its one-step expansion process, which is expected to minimize potential damage to biological architectures compared to multi-step expansions.

In the revised manuscript, the authors have addressed some of the major concerns raised in the initial submission, such as isotropic expansion and gel stability, by performing new experiments. However, the new images of nuclear pores and existing microtubules were not convincing in addressing these concerns. Instead, these images suggest that 20ExM may cause serious distortions in biological architectures, limiting its applications. The mouse brain images at low magnifications are beautiful, indicating that large-scale structures are likely well preserved. However, the primary users of 20ExM will be studying nanometer-scale structures, not just ~100 nm structures, which can already be studied with the original or existing ExM methods. Authors must provide strong evidence that nano-scale structures are preserved using 20ExM before publication in a leading methods journal, Nature Methods.

We thank the Reviewer for their remarks. Regarding the Reviewer's two major concerns (preservation of nanoscale biological architecture, and distortion in axial structures), it is important to note that certain characterizations by the Reviewer stand in contrast to the published literature, and other concerns are easily addressed by thinking about the nature of the experiment at hand.

Major concern

I understand and appreciate the effort to visualize nuclear pores as an additional nanoscale cellular ruler alongside microtubules. However, I have some concerns regarding the quality of the nuclear pore images. They appear unconvincing, making it difficult for readers to accept the quantifications. Many strong Nup96 puncta outside the circle marks are randomly distributed, and the nuclear pores marked as circles by authors do not resemble known nuclear pores. These images, along with some microtubule images, raise concerns about the preservation of nanoscale biological structures using 20ExM, potentially due to the 20-fold physical expansion.

The Reviewer argues that our images are unconvincing, due to the following two reasons:

1. "Many strong Nup96 puncta outside the circle marks are randomly distributed, and"
2. "the nuclear pores marked as circles by authors do not resemble known nuclear pores."

Response to Reason 1: As indicated in the Ext. Fig. 5 legend, we circled some nuclear pore complexes, but not all nuclear pores in the figure. Many puncta outside of the circles are likely part of other nuclear pore complexes. Importantly, in an earlier state-of-the-art ExM study visualizing nuclear pore complexes, even with specialized fixation, purification, and staining methods designed to optimize nuclear pore appearance, the investigators often observed single puncta that do not form a ring, such as in the image on the right of Supp. Fig. 5a (added to the new version of the manuscript), taken from the aforementioned state-of-the-art previous ExM paper that looks at nuclear pores. These puncta could arise from non-specific staining, or from a lack of staining against a fraction of the Nup96 proteins within a given pore.

Response to Reason 2: We disagree with the Reviewer's characterization that nuclear pores marked with circles do not resemble known nuclear pores. As shown in Supp. Fig. 5 comparing our NPC images (a, left; b, top row) with recently published state-of-the-art iU-ExM images (a, right; b, bottom row), achieved with a similar staining strategy (using one antibody, against a fluorescent protein attached to Nup96), but with additional methods designed to optimize nuclear pore appearance (e.g., nuclear extraction, PFA fixation at a different concentration), we observe highly similar appearance of NPCs.

In short, our data match state-of-the-art outcomes of previous ExM studies of the nuclear pore, that involve more elaborate expansion protocols. We note that our paper is not trying to exceed the performance of previous 20x expansion protocols, but only to create the most efficient way of achieving that performance - replacing laborious iterative expansion procedures, with a single, simple, step. The close match of our work, to a state-of-the-art precedent, achieves that goal.

We have now updated the manuscript to include this side-by-side comparison (in the new Supp Fig 5).

We tried to highlight these facts: as indicated in our previous **Response to Reviewers** (and associated Figure Legend), "We noticed some single-puncta signals that did not participate in a ring. We did not use a special NPC preparation strategy, as is common for microtubules. More specialized fixation methods, such as permeabilization with detergent prior to fixation, cryofixation with methanol, or extracting nuclei from intact cells prior to staining and expansion, might in principle further improve staining quality. Our goal in the current study was not to study NPCs, but rather to validate the resolution and gel-contributed error of 20ExM with NPCs. Furthermore, our goal was not to better earlier iterative methods like iExM or ExR, but rather to show that such performance could be achieved in a single step. Since our current protocol was sufficient for these purposes, we did not pursue further optimization." These specialized methods were developed to selectively improve the appearance of NPCs, whereas we used a standard PFA fixation method, as used most commonly by users of ExM. Indeed, the method of extracting nuclei out of cells through centrifugation prior to fixation, staining, and expansion, as used in the iU-ExM study, does not resemble a typical ExM user's application. The method described in *Nat. Methods* 16, 1045–1053 (2019), could in principle improve the appearance of nuclear pores, by reducing single-puncta signals that do not participate in rings, but is highly

specialized: the permeabilization protocol involves pre-fixation with 2.4% PFA, plasma membrane permeabilization with digitonin, a first round of staining, fixation with 3% PFA, nuclear envelope permeabilization with Triton X-100, a second round of staining, and final washes.

The Reviewer also mentioned that some microtubule images raised concerns, but did not elaborate. Our microtubule images are consistent with previous studies in both appearance, and measured thickness (Supp. Fig. 5g,h, added to the new version of the manuscript). Unlike NPCs, we have previous experience in preserving ultrastructure (with specialized fixation) of microtubules. Our microtubule images indicate that with such ultrastructure preparation, 20ExM produces highly similar images as previous iterative expansion methods and STORM, as shown in Supp. Fig. 5g,h.

We have now updated the manuscript to include this side-by-side comparison (in the new Supp Fig 5).

Additionally, in the new Supplementary Figure 3, the authors demonstrated isotropic expansion in the Z-axis and calculated approximately 5% distortion. However, the images appear significantly different before and after ExM, making it challenging to agree with the 5% distortion claim. There are many new signals in the post-20ExM images, and shapes differ significantly, even in regions where both images show signals. Again, these results raise concerns that 20ExM induces significant distortion in axial structures. The new mitochondria and brain images, while demonstrating micron/submicron scale preservation, do not convincingly prove the preservation of nanoscale architecture, which is a primary focus for most 20ExM users. Artificial distortion due to physical expansion and/or protein digestion remains a significant concern in ExM, especially with expansions larger than the original 4-fold.

Regarding z-axis distortion, the Reviewer pointed out that there are differences between pre- and post-expansion images, due to the following two reasons:

1. "There are many new signals in the post-20ExM images, and"
2. "shapes differ significantly, even in regions where both images show signals."

However, the Reviewer's comment that these differences indicate axial distortion introduced by 20ExM does not consider our experimental paradigm. We encountered two challenges when performing z-axis distortion analysis, including:

1. While we made our best attempts to ensure samples are in the same orientation during confocal imaging before and after expansion, slight differences (less than 10° rotational difference, based on our visual examination) occurred. While miniscule, the slight rotation of the sample during confocal imaging before and after expansion resulted in a slight angle difference in the xz and yz plane that we select in the same sample's pre- and post-expansion images. We found this challenging to correct computationally. Indeed, in almost all ExM papers that compare pre- and post-expansion images, there is some difference between the appearance of the two images, due to this aspect of the imaging process. It's simply a reality of imaging the same sample twice.

2. Because we use relatively thick brain tissue slices (50 μm thick), we observed a decrease in signal intensity at high-z positions (i.e., further away from the imaging lens, or the right side of the “YZ plane” images in Supp. Fig. 3d), due to scattering of light by lipids, whereas expanded (and therefore cleared) samples do not suffer from the same signal decrease. This resulted in apparent differences at high-z positions between pre- and post-expansion images.

Regarding Reason 1: the new signals in post-20ExM images are due to Challenge 2, that we described above. For example, the red-boxed area in the post-20ExM image is highly visible, whereas the boxed area in the pre-20ExM image only contains a faint signal (Supp. Fig. 3d, newly added to the current version of the manuscript). That is because this boxed area is at a deep z-depth, as far as confocal imaging is concerned. When we examine the same area in the xy-plane image at this z position, we observe the same neuron in pre and post-20ExM images. But, consistent with Challenge 2, in the “Pre XY plane” image, there is less signal than in the “Post XY plane” image. Consistent with Challenge 2, all of the reviewer’s claimed “new signals in the post-20ExM images” are at high-z positions, where pre-20ExM images have low signal intensity due to confocal limitations in scattering tissue.

Regarding Reason 2: the overall shape of soma and dendrites are similar between pre- and post-20ExM images. The slight difference in shape is likely due to Challenge 1.

The authors must provide validated structures that ensure nanoscale structural preservation, which is the principal concept of 20ExM for a broad audience. This might be achievable using centrosome 9-cartwheel structures, among others. If antibody size is a concern, labeling can be performed by expressing tag-conjugated proteins such as GFP (enhanced by commercial nanobodies for GFP to minimize extra size), SNAP, etc.

Regarding the Reviewer’s suggestion to use centrosome 9-cartwheel structures, among others, as “validated structures that ensure nanoscale structural preservation,” it is not clear to us how a structure like the centrosome can offer additional information beyond what we’ve validated with microtubule and NPCs. Centrosomes consist of nine microtubule structures, which are 1-2 μm away from each other (iU-ExM, Fig. 5d, *Nat. Commun.* 14, 7893 (2023); Magnify, Fig. 6, *Nat. Biotechnol.* 41, 858–869 (2023)), significantly above the 100 nm threshold for evaluating nanoscale biological structures. Yet, both microtubules (~60 nm peak-to-peak, as here stained) and NPCs (~40 nm corner-to-corner) structures are smaller than 100 nm and already count as nanoscale biological structures.

In summary, we think that we have shown our protocol to be the equal in performance to previous 20x protocols. We did not intend to exceed their performance, only to make the protocol practical for everyday biology.

Minor concern

The authors addressed the expected signal reductions due to 20ExM in a step-by-step response. However, this significant limitation could severely restrict its applications, as it will not work with typical primary-secondary antibody staining. While this limitation does not diminish the impact of the technology, the authors must clearly and extensively discuss these points in the Main text. Specifically, they should address the expected signal reductions, the necessity for tertiary antibody staining, the potential limitations, and possible ways to overcome these technical barriers. These aspects are crucial for readers and future users. time will need further adjustments, or do they expect that the 4 uL /20-hour 20 uL / 4-hour benchmarks will serve a diverse spectrum of tissues, or do you expect modifications of these ratios to ensure maximum expansion?

We thank the Reviewer for raising these concerns regarding signal amplification and gelation condition. Regarding signal amplification, we have added text in the Discussion section with more extensive information in Supp. Note 1. We disagree that the tertiary antibody staining severely restricts its application, since multi-color staining can be achieved in the same fashion as primary-secondary antibody staining, with orthogonal antibody sets.

Regarding gelation condition, we have responded in previous Response to Reviewers, Author's Response 19 and Main Text: "we recommend the standard gelation condition for all tissues, at least as a starting point (very complex tissues like bone and cartilage, or very large samples like entire mammalian brains, may of course require further tuning), but tissues with challenging mechanical properties may require harsher softening methods than heat/detergent treatment, such as enzymatic methods, many of which have already been validated and published by us and by others."